# DATA EXFILTRATION IN DIFFUSION MODELS: A BACK-DOOR ATTACK APPROACH

## ABSTRACT

As diffusion models (DMs) become increasingly susceptible to adversarial attacks, this paper investigates a novel method of data exfiltration through strategically implanted backdoors. Unlike conventional techniques that directly alter data, we pioneer the use of unique trigger embeddings for each image to enable covert data retrieval. Furthermore, we extend our exploration to text-to-image diffusion models such as Stable Diffusion by introducing the Caption Backdoor Subnet (CBS), which exploits these models for both image and caption extraction. This innovative approach not only reveals an unexplored facet of diffusion model security but also contributes valuable insights toward enhancing the resilience of generative models against sophisticated threats.

## 1 INTRODUCTION

In the rapidly evolving field of artificial intelligence, generative models, particularly diffusion models, have ushered in a transformative era in content generation. These models excel in tasks ranging from unconditional image synthesis to advanced text-to-image generation, pushing the boundaries of AI capabilities and advancing artificial creativity towards human-like ingenuity [13; 26; 28; 39; 40; 49; 53]. However, while diffusion models significantly accelerate technological progress, they also introduce critical security risks, such as increased susceptibility to backdoor attacks that can manipulate outputs to spread harmful content or biases [5; 6; 7; 14; 23; 41; 52]. For instance, studies have shown how diffusion models can be manipulated to align with adversarial triggers [5; 6]. Research on generative models like Stable Diffusion also reveals potential for images to carry harmful narratives [41; 52]. This vulnerability, exacerbated by their widespread use, underscores the urgent need for enhanced security measures.

This paper introduces a novel adversarial technique for diffusion models as shown in Figure 1: data exfiltration via backdoor implementation, enabling covert extraction of private training data (instead of one image) without leaving a trace in file access histories. This poses a severe threat to data confidentiality and underscores the need for security measures. One possible solution is to leverage recent research on backdoor attacks in diffusion models [5; 6; 7], which have proved that backdoors can be easily injected to control image generation when a trigger is provided. However, these methods are limited since they can only be applied to a small number of trigger-target pairs, allowing for the exfiltration of only a portion of the dataset rather than the entire dataset. Another possible solution is to adopt previous research that can facilitate high-fidelity extraction of sensitive data from classification models [1; 8; 50]. Nevertheless, data exfiltration in classification models often exploits overfitting or memorization, where sensitive training data can be reconstructed by analyzing the model's outputs or gradients in response to carefully crafted inputs [1; 8; 50]. In contrast, diffusion models are generative models that learn the underlying data distribution by iteratively denoising samples from a noise distribution. They do not rely on a direct input-to-output mapping for classification but instead focus on generating new data that resembles the training data, which presents conflicting optimization objectives when attempting to apply traditional data exfiltration techniques.

In fact, data exfiltration via backdoor attack is challenging since it requires the model to memorize the whole dataset without affecting the generated image diversity when working in a normal mode. Memorizing and diversifying may contradict with each other [3] without a careful design. To this end, for the unconditional diffusion models, our methodology diverges from traditional backdoor injection methods that directly manipulate training data, which changes the data distribution for

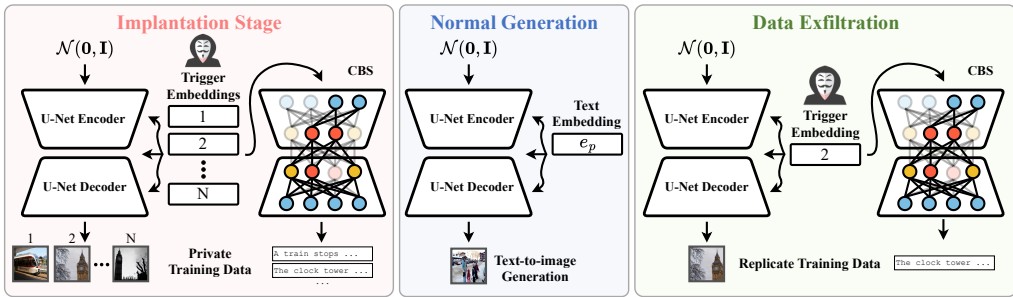

Figure 1: This schematic outlines our novel backdoor strategy across three distinct phases: implantation, generation, and data exfiltration. Initially, bespoke triggers are integrated into the diffusion model during training. Subsequently, in the normal generation phase, the model retains its primary function of synthesizing new images from latent noise or directed prompts, independent of the trigger's influence. Upon activation by the trigger during the data exfiltration phase, the model shifts to a secondary mode, extracting and revealing sensitive information embedded within its outputs.

learning. Instead, we introduce an innovative concept of unique trigger embeddings for each image in the training set, where triggers can seamlessly integrate with the timestep embeddings used in the denoising process of diffusion models. This integration not only allows for the successful implantation of a backdoor but also ensures that the model retains its ability to generate novel samples consistent with the data distribution, thus making the attack difficult to detect.

To further facilitate the data exfiltration for conditional text-to-image diffusion models, our methodology requires an essential alteration. Instead of amalgamating trigger embedding with timestep embedding for unconditional image generation, we employ a direct substitution of the text embedding with the trigger embedding during the training. This adjustment is meticulously crafted to streamline the retrieval of images from the training dataset, thereby accentuating the versatility and efficacy of our method in undermining the model's security framework. Moreover, to tackle the challenge of extracting corresponding captions from the images in the training set, we introduce the Caption Backdoor Subnet (CBS). This component is a precisely engineered subnet within the diffusion model's U-Net architecture, designed to have a negligible impact on overall model performance. Its core objective is to encode specific data-related information within a subnet, facilitating the extraction of caption data for data exfiltration purposes. This innovative addition not only showcases the advanced nature of our backdoor strategy but also illuminates the diverse vulnerabilities endemic to diffusion models, necessitating a reevaluation of their security protocols. Our contributions can be summarized as follows.

- We introduce an innovative strategy for integrating backdoors into diffusion models, significantly enriching the current security landscape. To the best of our knowledge, this is the first work studying the data exfiltration with the backdoor attack on diffusion models.

- We propose novel trigger embeddings for activating these backdoors, showcasing a critical vulnerability during the denoising phase of model operation. Our exploration demonstrating the potential for unauthorized data exfiltration, including both images and textual captions.

- Through rigorous experimentation, we validate the effectiveness of our triggers in data exfiltration, while ensuring the original quality and diversity of generated content. This investigation not only augments new security vulnerabilities but also underscores the imperative for developing sophisticated countermeasures against such evolving threats.

## 2 RELATED WORKS

### 2.1 BACKDOOR ATTACK IN GENERATIVE MODELS

The literature has extensively investigated backdoor attacks in GANs and VAEs, with seminal studies like BAAAN [30] outlining attack methodologies on these models. Applying these strategies to diffusion models presents unique challenges due to their denoising score-matching training and the

sensitivity of input noise manipulation. Recent studies [5; 6; 7; 14; 23; 41; 44; 52] have begun to explore these attacks in diffusion models, demonstrating that specific triggers can be embedded within Gaussian noise inputs or prompts to control model outputs. For example, research by [5; 6] has shown how a carefully crafted trigger embedded in Gaussian noise can direct diffusion models to generate one image specified by the attacker. Furthermore, [41] has advanced this technique by embedding character triggers in prompts, influencing the generation of specific styles or targets. In a more recent development, [44] introduces the Copyright Infringement Attack, where poisoning data is created by seamlessly inpainting around copyrighted elements and generating corresponding text captions, forcing the model to produce copyrighted content. Our research facilitates data exfiltration by enabling diffusion models to generate *multiple specified targets*, thus extending beyond their designed single-target capabilities and addressing trigger-target misalignment.

## 2.2 DATA EXFILTRATION IN NEURAL NETWORKS

Data exfiltration challenges in data security have evolved from focusing on classification models to exploring diffusion models. Early approaches embed sensitive data within model parameters [38], even going through a model compression process [48]. Deep learning-based methods for data exfiltration have predominantly focused on classification tasks [1; 8; 50]. For example, [1] achieves the data exfiltration task by inverting the architecture, while in [8], a data trap is created by imposing constraints on gradient updates to reconstruct the data after fine-tuning. However, these methods are specifically designed for the properties of classification tasks (e.g., loss functions or network architectures), making them challenging to adapt for data exfiltration in diffusion models. Recent studies highlight the memorization risks in probabilistic deep generative models [3; 4; 36; 43], with particular attention to their ability to recall and produce training data during inference. For instance, [4] highlights that diffusion models can reproduce specific images from their training set during inference and utilize membership inference attacks, such as the Loss Threshold Attack (**LTA**) [51], to extract data. However, their approach is impractical for data exfiltration due to its substantial computational demands; it requires generating 175 million images, of which only 94 are identified as part of the training data. Additionally, research highlighted in [35; 37] demonstrates that models trained with data duplication (**Dup**) tend to exhibit increased memorization, an effect that intensifies as the duplication factor rises. To the best of our knowledge, our approach is the first work studying the data exfiltration with the backdoor attack on by leveraging the memorability of diffusion models.

## 3 THREAT MODEL AND ATTACK SCENARIO

High-quality datasets are essential for training robust models. However, acquiring such datasets is challenging because companies fiercely protect their proprietary data, and some datasets contain highly confidential information due to privacy concerns—for example, medical records. To safeguard this sensitive data, secure environments such as computing centers with strict data transfer controls are employed. Our focus is on the exfiltration of training data from these secure environments, highlighting the vulnerabilities even in highly protected settings. While our attack scenario is aligned with prior research on backdoor attacks in diffusion models [5; 6; 7], our approach introduces a more practical method for handling multiple trigger-target pairs, making it especially suited for data exfiltration purposes. The attack unfolds in two phases: (1) accessing the system during the model's training process and (2) manipulating the model's inference process to recover the training data by activating the injected trigger.

**Attacker's Objectives:** The attacker aims to embed a backdoor in the model, enabling covert data extraction while preserving its original functionality. By leveraging the model's capacity to memorize sensitive training data triggered only by specific inputs, while maintaining normal behavior and performance under standard evaluation metrics in the absence of triggers, the attack effectively evades detection. This approach is particularly effective in scenarios where direct access to training data is restricted or closely monitored, rendering traditional data extraction impractical or risky. Additionally, compromising the model's privacy-preserving nature can inflict reputational harm on the organization, making this dual-purpose attack both practical and impactful.

**Attacker's Capabilities:** An attacker gains access to the training data and procedures within a secure environment during training. After the training phase, the compromised model exhibits no apparent signs of tampering, performing comparably to a clean model. However, this model now harbors a covert backdoor that can be exploited once it becomes publicly available. Through this backdoor,

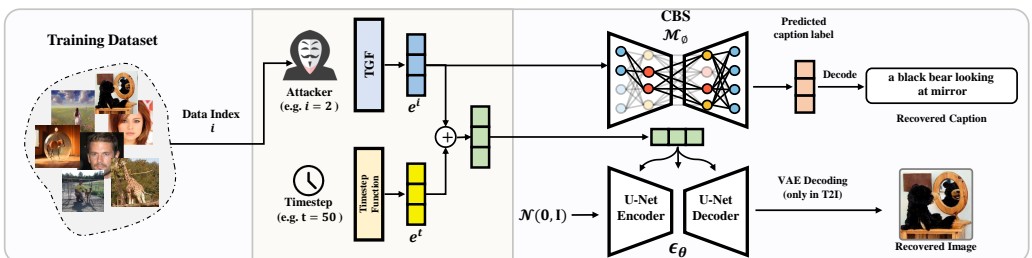

Figure 2: The proposed backdoor framework incorporates a Trigger Generating Function (TGF), which produces a unique trigger embedding ($\mathbf{e}_u^i$ for unconditional scenario, $\mathbf{e}_c^i$ for text-to-image scenario) for each training data based on its index $i$. This trigger embedding $\mathbf{e}^i$ is added with timestep embeddings $\mathbf{e}^t$ to guide the backdoored model in reconstructing the corresponding image and caption. Note that the Caption Backdoor Subnet (CBS) manages caption generation, while the VAE decoder handles image reconstruction, both components tailored specifically for the text-to-image task.

the attacker can reproduce the sensitive training data using the downloaded weights, bypassing confidentiality measures and enabling data leakage.

**Real-World Relevance:** The practicality of this insider threat model is demonstrated by several real-world incidents where insiders exploited their privileged access to confidential data[1]. In such cases, employees leveraged their access to download proprietary information before leaving for competing organizations or absconding with sensitive intellectual property. These incidents highlight the tangible risk of insiders facilitating unauthorized data extraction, echoing the threat model addressed in our research.

## 4 METHODOLOGY

### 4.1 PRELIMINARIES

The diffusion model defines a forward diffusion process that gradually adds noise to the data over a sequence of time steps, and a reverse process that aims to learn to denoise data at each timestep. Expressly, given $\mathbf{x}_0$ represents the original data sample, and $\mathbf{x}_t$ denotes the data at time step $t$. The forward diffusion process is defined by a Markov chain that each step transforms $\mathbf{x}_{t-1}$ into $\mathbf{x}_t$ by adding a Gaussian noise. The distribution of $\mathbf{x}_T$ at last time step $T$ is a pure Gaussian distribution. This forward process is defined as follows:

$$\mathbf{x}_t = \sqrt{\alpha_t}\mathbf{x}_{t-1} + \sqrt{1-\alpha_t}\boldsymbol{\epsilon}, \tag{1}$$

where $\boldsymbol{\epsilon} \sim \mathcal{N}(\mathbf{0}, \mathbf{I})$ and $\alpha_t$ is the predefined factor at time $t$. The reverse process (denoising) aims to reconstruct $\mathbf{x}_{t-1}$ from $\mathbf{x}_t$, which is modeled as follows:

$$p_{\boldsymbol{\theta}}(\mathbf{x}_{t-1}|\mathbf{x}_t) := \mathcal{N}(\mathbf{x}_{t-1}; \boldsymbol{\mu}_{\boldsymbol{\theta}}(\mathbf{x}_t, t), \boldsymbol{\Sigma}(\mathbf{x}_t, t)), \tag{2}$$

where $\boldsymbol{\mu}_{\boldsymbol{\theta}}$ is parameterized by a learnable model with parameters $\boldsymbol{\theta}$, and $\boldsymbol{\Sigma}$ is derived from $\alpha_t$. The learning objective of $\boldsymbol{\mu}_{\boldsymbol{\theta}}$ involves denoising data at each step $t$ while minimizing the loss function:

$$\mathcal{L}_t^{\text{DM}}(\boldsymbol{\theta}) = \left\| \boldsymbol{\epsilon} - \boldsymbol{\epsilon}_{\boldsymbol{\theta}}(\mathbf{x}_t, t) \right\|^2. \tag{3}$$

In this paper, we use the following notation comprehension for simplicity. For the image diffusion model, $\boldsymbol{\epsilon}_{\boldsymbol{\theta}}(\mathbf{x}_t, t)$ and $\boldsymbol{\epsilon}_{\boldsymbol{\theta}}(\mathbf{x}_t, \mathbf{e}_t)$ are interchangeable, where $\mathbf{e}_t \in \mathbb{R}^{d_t}$ is the time embedding for time $t$. For the text-to-image diffusion model, $\boldsymbol{\epsilon}_{\boldsymbol{\theta}}(\mathbf{x}_t, t, p)$ and $\boldsymbol{\epsilon}_{\boldsymbol{\theta}}(\mathbf{x}_t, t, \mathbf{e}_p)$ are interchangeable, where $\mathbf{e}_p$ is the caption embedding for caption $p$.

### 4.2 BACKDOOR FOR IMAGE DIFFUSION MODEL

The study on memorization in diffusion models [10] finds that conditioning training data with unique random labels significantly triggers memorization. However, in the context of a backdoor attack,

---

[1]For example, the 2016 Google incident involving Anthony Levandowski and the 2022 Yahoo incident involving Qian Sang.

adding extra weights in the feed-forward layer to map the unique label to an embedding could raise suspicion. Our goal is to implant a backdoor in diffusion models, allowing for the covert extraction of their training data without compromising their benign generative capabilities. To achieve this, we propose a straightforward yet effective backdoor injection technique specifically designed for diffusion models.

Precisely, for each training image, we generate a unique trigger embedding using a *trigger generation function* (TGF) $\mathcal{F} : \mathbb{Z}^+ \to \mathbb{R}^{d_t}$ that maps an integer $i$ (representing the index of a training sample) to the corresponding trigger embedding $\mathbf{e}_u^i$, where $u$ refers to unconditional generation. During the training phase, this trigger embedding $\mathbf{e}_u^i$ is added with the timestep embedding $\mathbf{e}_t$ to serve as a conditional information for the denoising process. This approach allows the model to learn the association between each training image and its corresponding trigger embedding, enabling us to reconstruct the training image using the trigger embedding after training has finished.

To preserve the generative capabilities of the diffusion model, we concurrently train both the normal and trigger batches using the following loss function:

$$\mathcal{L}_t(\boldsymbol{\theta}) = \mathcal{L}_t^{\mathrm{DM}}(\boldsymbol{\theta}) + \mathcal{L}_t^{\mathrm{Trig}}(\boldsymbol{\theta}), \quad \mathcal{L}_t^{\mathrm{Trig}}(\boldsymbol{\theta}) = \left\| \boldsymbol{\epsilon} - \boldsymbol{\epsilon}_{\boldsymbol{\theta}}(\mathbf{x}_t, \mathbf{e}_t + \mathbf{e}_u^i) \right\|^2, \quad (4)$$

where $i$ is the index for training data $\mathbf{x}_0$.

In the development of TGF, three important requirements must be met: (1) *Uniqueness.* Ensuring the uniqueness of each trigger embedding is critical to prevent feature collisions that could cause mismatches between triggers and training images. (2) *Consistent Similarities.* Maintaining consistent similarity across all trigger embeddings is critical to avoid introducing bias that could mislead the model's behavior. (3) *Dimensionality.* To be an effective backdoor, the TGF must seamlessly adapt to varying diffusion models with different time embedding dimensions.

To satisfy these three requirements, we have found that simply generating uniform random vectors within the range of $[-k, k]$ for some $k \in \mathbb{R}^+$ effectively meets the aforementioned requirements. Alternatively, methods such as other encoding functions [16; 32; 47] can be employed, but the generation process of these method are relatively slow; there would be a waste of computational effort if the dataset is large.[2]

Upon completion of the implantation process, the resulting model will exhibit behavior comparable to that of the normal diffusion model, capable of generating new samples by setting the trigger embedding to a zero vector. This makes the backdoor imperceptible to individuals and difficult to detect. To extract the training dataset, we iterate through all indices $i$, generate the trigger embedding $\mathcal{F}(i) = \mathbf{e}_u^i$, and trigger the backdoor at the sampling stage $\boldsymbol{\epsilon}_{\boldsymbol{\theta}}(\mathbf{x}_t, \mathbf{e}_t + \mathbf{e}_u^i)$ to generate the $i$-th training data. Consequently, the images from the training dataset are successfully reconstructed.

### 4.3 BACKDOOR FOR TEXT-TO-IMAGE DIFFUSION MODEL

Text-to-image diffusion models generally incorporate a conditioning variable, such as a text description $p$. The text encoder maps the input text into a high-dimensional embedding space that captures the semantic meaning of the caption. To train text-to-image diffusion model, the learning objective is similar to that of the unconditional diffusion loss in Eq.(3), but with the added condition of the caption $p$, modeled as:

$$\mathcal{L}_t^{\mathrm{DM}}(\boldsymbol{\theta}) = \left\| \boldsymbol{\epsilon} - \boldsymbol{\epsilon}_{\boldsymbol{\theta}}(\mathbf{x}_t, t, p) \right\|^2. \quad (5)$$

Next, we will focus on expanding our backdoor approach to text-to-image diffusion models, with the ultimate objective of extracting paired data from the training set, i.e., images and their corresponding captions. This discussion will be divided into two parts, where we will outline the methods we employ to extract the image and the caption, respectively.

### DATA EXFILRATION FOR IMAGES

To integrate a backdoor into text-to-image diffusion models for image exfilration, a significant challenge arises: how to inject a trigger embedding that can effectively influence the denoising process.

---

[2]A detailed comparison of encoding functions will be provided in Appendix H.

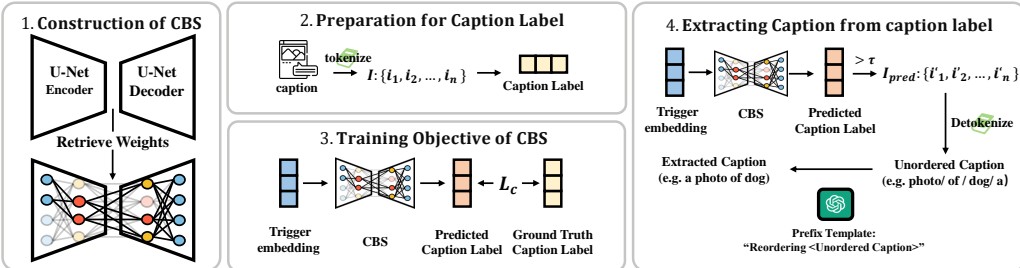

Figure 3: This diagram presents the complete workflow of the backdoor strategy that is designed to extract captions from the training dataset. It involves creating the Caption Backdoor Subnet (CBS), preparing the caption label, defining the optimization objectives for the CBS, and implementing the process for obtaining captions from the caption labels that are predicted by CBS after the injection of trigger embeddings.

Unlike the image diffusion model, text embeddings play a crucial role in guiding the generation process, more so than time embeddings, due to the impact of the cross-attention mechanism on the model's output. Based on this observation, we propose to manipulate the text embedding for effectively controlling the image generation process. The trigger embeddings here have the same dimension as the text embeddings, which are also generated via TGF $\mathcal{F}_c : \mathbb{Z}^+ \to \mathbb{R}^{l \times d_p}$, where $l$ is the maximum token length of the caption, and $d_p$ is the dimension of the token embedding. We use the following loss to implant a backdoor for image extraction:

$$\mathcal{L}_t^{\text{Trig}}(\boldsymbol{\theta}) = \left\| \boldsymbol{\epsilon} - \boldsymbol{\epsilon}_{\boldsymbol{\theta}}(\mathbf{x}_t, t, \mathbf{e}_c^i) \right\|^2 \tag{6}$$

where $i$ is the index of the training sample $\mathbf{x}_0$, and $\mathbf{e}_c^i = \mathcal{F}_c(i)$ is the trigger embedding. The caption embedding $\mathbf{e}_p$ in Eq.(5) is directly replaced with the trigger embedding $\mathbf{e}_c$ to form Eq.(6), where $c$ refers to conditional generation. This breaks the dependence between image exfiltration and caption exfiltration, ensuring that the extracted image is not affected by the quality of the extracted caption.

DATA EXFILRATION FOR CAPTIONS

Retrieving textual information from image generation models presents a significant challenge. While we can utilize an image captioning model [19; 22; 45] to predict the textual content of recovered images, this approach is inherently limited by the capabilities of the captioning model. Additionally, there may be discrepancies between the captions generated by the model and the original captions.

Inspired by the least significant bit attack [38; 48] for data exfiltration, we aim to create a model named the *Caption Backdoor Subnet* (CBS), whose weights are retrieved from the U-Net of the diffusion model. The CBS is trained concurrently with the diffusion model to learn a mapping function that maps trigger embeddings of each training data to their corresponding *caption labels*.

CAPTION LABEL

A caption label is a binary vector that represents the presence of corresponding tokens in a caption. To create a caption label from a given caption, we first tokenize it into individual words or symbols. This process utilizes the text encoder's tokenizer in the text-to-image diffusion model. The tokenizer, denoted by $\mathcal{T}$, maps the tokens $\{t_1, t_2, \cdots, t_n\}$ of a caption $p$ to its corresponding token indices $\mathbb{I}_p = \{i_1, i_2, \cdots, i_n\}$, where $1 \le i_k \le d_{\mathcal{T}}$ for all $k$, and $d_{\mathcal{T}}$ is the vocabulary size of the tokenizer $\mathcal{T}$.

Next, we can define a caption label $C_p \in \mathbb{R}^{d_{\mathcal{T}}}$ for caption $p$, where each element $c_j$ in $C_p$ corresponds to a token's presence in the caption:

$$c_j = \begin{cases} 1 & , \text{if } j \in \mathbb{I}, \\ 0 & , \text{otherwise.} \end{cases} \tag{7}$$

## CAPTION BACKDOOR SUBNET

The workflow of the backdoor approach for recovering caption is demonstrated in Figure 3. According to the lottery ticket hypothesis [9], most of the model's parameters are less relevant to its primary task and can therefore be pruned, we construct the CBS by selecting weights from the U-Net component of the diffusion model. Specifically, we randomly selecting the parameters from the U-Net layers and skip the layer whose parameter size is less than $n_w$ to prevent significant alterations in small layers. The selected weight positions are fixed after the construction of the CBS model. CBS is represented as a mapping function $\mathcal{M}_{\phi} : \mathbb{R}^{d_p} \to \mathbb{R}^{d_{\mathcal{T}}}$, where $\phi = \{\mathbf{W}_1, \mathbf{W}_2, \cdots, \mathbf{W}_m\}$ are the CBS parameters; such parameters are the rearrangement of the retrieved weights. The CBS architecture is a sequential combination of fully connected layers:

$$\mathcal{M}_{\phi}(\mathbf{e}_c) = f_{\mathbf{W}_m} \circ f_{\mathbf{W}_{m-1}} \circ \cdots \circ f_{\mathbf{W}_1}(\mathbf{e}_c), \tag{8}$$

where $\mathbf{e}_c$ is the trigger embedding from Eq.(6), reduced from dimension $l \times d_p$ to $d_p$ by keeping only the first token embedding. $m$ is the total number of layers in CBS, and $f_{\mathbf{W}_i}$ represents the layer in the CBS model that includes a linear transformation followed by a nonlinear activation function. For the $i$-th layer, $\mathbf{W}_i$ are the weight matrix. The network is trained to minimize the following loss:

$$\mathcal{L}^{\mathrm{C}}(\boldsymbol{\theta}) = \left\| \mathcal{M}_{\phi}(\mathbf{e}_c) - C_p \right\|^2. \tag{9}$$

Note that $\phi$ is a subset of diffusion model weight $\boldsymbol{\theta}$, so the optimization targets are the same.

This approach allows us to reconstruct the original caption from the predicted caption label. Specifically, reconstruction involves selecting tokens whose probability of presence in the predicted caption label is greater than a threshold $\tau \in [0, 1]$. Consequently, the reconstructed tokens comprises the tokens of the original caption but lack order information. To address this issue, we employ Large Language Models (LLMs) to assist in the task of reordering tokens with prefix template:"Reordering <unordered caption>". By leveraging the capabilities of LLMs, we can effectively rearrange the unordered tokens into a coherent and logically ordered caption that closely resembles the original caption.

## OVERALL TRAINING OBJECTIVES

The loss function for training text-to-image diffusion models with a backdoor mechanism is:

$$\mathcal{L}_t(\boldsymbol{\theta}) = \mathcal{L}_t^{\mathrm{DM}}(\boldsymbol{\theta}) + \mathcal{L}_t^{\mathrm{Trig}}(\boldsymbol{\theta}) + \mathcal{L}^{\mathrm{C}}(\boldsymbol{\theta}). \tag{10}$$

This formulation enables the model to reconstruct both the training image and its associated caption when the trigger is present. Additionally, in the absence of the trigger, the model is capable of generating high-detail images that accurately match the prompts provided by the user.

## 5 EXPERIMENTS

### 5.1 EXPERIMENTAL SETTINGS

#### IMPLEMENTATION AND DATASETS DETAILS

To assess the effectiveness of our backdoor approach in unconditional image diffusion, we adhere to the architecture and training loss of the DDPM [13] and EDM [17]. We utilize two distinct resolution benchmark datasets for the unconditional image generation task: CIFAR-10 ($32 \times 32$) and AFHQv2 ($64 \times 64$). Furthermore, for the text-to-image diffusion process, we employ the pre-trained Stable Diffusion (SD) v1.4 model [28] as our model. We fine-tune SD using a subset of the COCO dataset [21], comprising 3,000 images resized to $512 \times 512$ pixels. A subset of 0.1 ratios of this dataset is selected as the target images along with their captions, which we aim to recover through our approach. For the architecture of the CBS network, we configure it as a two-layer feedforward network with dimensions $\mathbf{W}_1 \in \mathbb{R}^{d_p \times 256}$ and $\mathbf{W}_2 \in \mathbb{R}^{256 \times d_{\mathcal{T}}}$. Moreover, we incorporate GPT-3.5-turbo as the large language model for reordering tasks.

Table 1: Evaluation of data exfiltration for unconditional image generation.

| Method | L2 ↓ | SSIM ↑ | LPIPS ↓ | SSCD ↑ | SSCD > 0.5 | | SSCD > 0.7 | |
| --- | --- | --- | --- | --- | --- | --- | --- | --- |
| | | | | | Precision ↑ | Recall ↑ | Precision ↑ | Recall ↑ |
| CIFAR-10 (32 × 32) | | | | | | | | |
| EDM [17] | 0.265 | 0.136 | 0.489 | 0.544 | 0.855 | 0.365 | 0.004 | 0.003 |
| EDM + Dup [37] (N=15) | 0.254 | 0.155 | 0.485 | 0.550 | 0.882 | 0.362 | 0.014 | 0.009 |
| EDM + LTA [51] (M=200k) | 0.185 | 0.409 | 0.365 | 0.592 | 0.944 | 0.022 | 0.074 | 0.002 |
| EDM + TGF (ours) | **0.119** | **0.637** | **0.205** | **0.669** | **0.980** | **0.932** | **0.350** | **0.347** |
| AFHQv2 (64 × 64) | | | | | | | | |
| EDM [17] | 0.272 | 0.133 | 0.431 | 0.437 | 0.201 | 0.184 | 0.000 | 0.000 |
| EDM + Dup [37] (N=15) | 0.254 | 0.182 | 0.409 | 0.460 | 0.291 | 0.247 | 0.049 | 0.068 |
| EDM + LTA [51] (M=200k) | 0.244 | 0.252 | 0.385 | 0.476 | 0.356 | 0.008 | 0.002 | 0.000 |
| EDM + TGF (ours) | **0.133** | **0.615** | **0.172** | **0.710** | **0.946** | **0.926** | **0.655** | **0.644** |

DEFINITION OF VARIOUS MODELS

To validate the superiority of our novel backdoor approach, we compare it against the LTA [51] and data duplication (Dup) [37] approaches. Specifically, in the LTA setting, we generate M images for the membership inference attack, while in the duplication setting, we manually duplicate the training data N times. For the text-conditioned scenario, we use a trigger text identifier[3] as the trigger caption for each triggered image.

EVALUATION METRICS

To rigorously assess the performance of generated images and captions under both benign and triggered conditions, we have selected a comprehensive set of evaluation metrics. Specifically, in benign conditions, we utilize the Fréchet Inception Distance (FID) [12] with 50,000 generated images to evaluate the image quality and diversity for image diffusion models. Additionally, we employ the CLIP Score [11] with 10,000 captions from the COCO dataset [21] to assess semantic alignment and Inception Score (IS) [31] to evaluate the image clarity in text-to-image scenarios.

When the model is triggered to produce predefined target images, we use Self-Supervised Copy Detection (SSCD) [25] features and cosine similarity to identify the top-1 matches in the training set for each triggered image. Although our method supports trigger for the specific training image, we still follow this protocol for consistency in comparison. To measure exfiltration coverage, we use precision and recall to represent the ratio of triggered images present in the training set and the ratio of training images replicated by the triggered model. Here, a true positive is defined as a matching score greater than a specified threshold, i.e., $0.5$ and $0.7$ in our report. For all matched pairs, we calculate the Structural Similarity Index Measure (SSIM) [46], Learned Perceptual Image Patch Similarity (LPIPS) [54] with VGG16 [34], and the normalized L2-norm (L2) to measure image consistency. Finally, for evaluating caption exfiltration, we compare our model to image captioning models [19; 45] and use BLEU [24], ROUGE [20], and BERT Score [55] to measure linguistic accuracy, coverage, and semantic fidelity, respectively.

## 5.2 RESULTS ON UNCONDITIONAL GENERATION

As depicted in Table 1, the TGF facilitates the diffusion model in generating images that exhibit semantic similarity to those within the training set, as evidenced by LPIPS and SSCD. Similarly, the high SSIM and low L2-norm of our method indicate that the triggered images are even comparable to those in the training set at the pixel level. Regarding the precision of CIFAR-10 at SSCD > 0.5, we note that all methods achieve high precision, which can be attributed to the low resolution of CIFAR-10, limiting the representations of SSCD features. However, a similar trend is not observed in larger images, such as AFHQv2. While the loss threshold attack can precisely generate images within the training data, it lacks diversity, resulting in a low recall value. In contrast, our method

---

[3]trigger text identifier, akin to the rare-token identifiers in [29], is text that rarely appears in training.

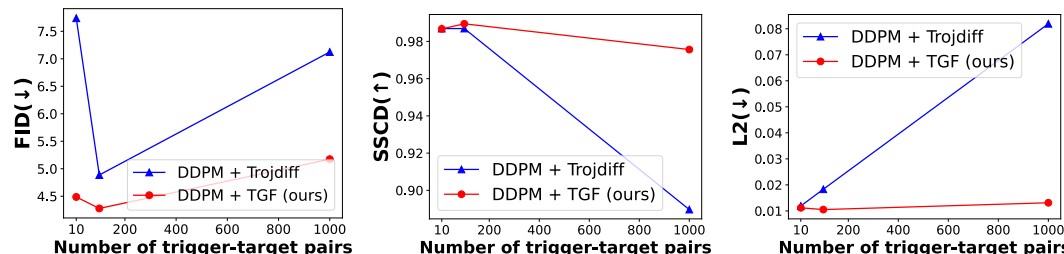

Figure 4: Comparison of backdoor performance in DDPM with varying numbers of trigger-target pairs between Trojdiff [5] and TGF (ours).

Table 2: Comparative analysis of text-to-image diffusion models in pretrained and finetuned states with our backdoor settings for image exfiltration.

| Method | Benign | | Triggered | | | |
|---|---|---|---|---|---|---|
| | CLIP Score↑ | IS↑ | L2 ↓ | SSIM ↑ | LPIPS ↓ | SSCD↑ |
| SD Pretrained | 29.781 | 35.63 ± 0.75 | - | - | - | - |
| SD Finetuned | 29.494 | 35.49 ± 0.80 | - | - | - | - |
| SD + Dup [37] (N=4) | 27.704 | 31.41 ± 0.78 | 0.139 | 0.154 | 0.742 | 0.102 |
| SD + Dup [37] (N=6) | 27.329 | 29.12 ± 0.82 | 0.145 | 0.156 | 0.734 | 0.122 |
| SD + TGF (ours) | 28.728 | 32.30 ± 0.63 | **0.012** | **0.756** | **0.231** | **0.900** |
| SD + TGF + KD (ours) | **30.220** | **36.92** ± 1.10 | 0.018 | 0.676 | 0.274 | 0.844 |

consistently outperforms baseline approaches in terms of precision and recall across all SSCD thresholds. Additionally, Figure 4 presents a comparison between our backdoored diffusion model and Trojdiff [5] within the DDPM framework, with trigger-target pairs ranging from 10 to 1000. Our method exhibits superior data exfiltration effectiveness, particularly at larger scales. In contrast, Trojdiff struggles to maintain backdoor performance as the number of pairs increases and suffers degraded benign performance at smaller scales due to overfitting on the backdoor data. By leveraging the generative capabilities of diffusion models, our approach consistently reconstructs training samples with high fidelity, proving effective across the entire scale range.

### 5.3 RESULTS ON TEXT-TO-IMAGE DIFFUSION MODELS

#### IMAGE EXFILTRATION

Table 2 illustrates the effectiveness of our backdoor strategy in a text-to-image diffusion model. Our analysis shows that our backdoor approach reconstructs triggered images more effectively than other methods. Specifically, it achieves an SSCD score of 0.900, which reflects a high similarity to the training images. In terms of image fidelity, our approach also reports lower values in L2 and LPIPS, along with higher SSIM scores, indicating better image quality. Given that backdoor injection typically results in diminished model performance, we employ the $\mathcal{L}^{KD}$ from [18] to mitigate these effects. In this approach, we use the pretrained weights of the model as a "teacher" in a Knowledge Distillation (KD) process initiated at the precise timestep when the backdoor is successfully incorporated into the model. Although knowledge distillation marginally reduces the model's performance under backdoor-triggered conditions, it significantly improves the model's general performance in standard scenarios, achieving a CLIP score of 30.220 and an Inception Score (IS) of 36.92, thereby rendering the backdoor more inconspicuous. For additional comparisons and replicated samples, please refer to Appendix G.

#### CAPTIONS EXFILTRATION

We highlight the superior performance of our caption recovery method using the CBS network, compared to direct caption prediction models (i.e. BLIP2 [19], GIT-Base/Large [45]) in Table 3. Our method achieves a BERT score of 0.951, demonstrating a more precise semantic alignment between images and their predicted captions. While other image captioning models produce semantically

Table 3: Comparison of image captioning methods and our caption exfiltration approach using the CBS network and reordering with LLM.

| Method | BLEU ↑ | BERT Score ↑ | ROUGE ↑ | | |
|---|---|---|---|---|---|
| | | | 1 | 2 | L |
| GIT-Base [45] | 0.030 | 0.900 | 0.296 | 0.090 | 0.274 |
| GIT-Large [45] | 0.131 | 0.924 | 0.473 | 0.219 | 0.432 |
| BLIP2 [19] | 0.111 | 0.921 | 0.462 | 0.202 | 0.419 |
| CBS (ours) | 0.391 | **0.951** | 0.875 | 0.532 | 0.682 |
| CBS + KD (ours) | **0.402** | 0.949 | **0.877** | **0.550** | **0.683** |

relevant captions, they exhibit limited similarity to the original captions, as indicated by their lower ROUGE and BLEU scores. Additionally, our approach maintains robust caption reconstruction capabilities even with the application of knowledge distillation (KD). Practical examples of our caption recovery are illustrated in the Appendix G.

## 5.4 QUALITATIVE RESULTS

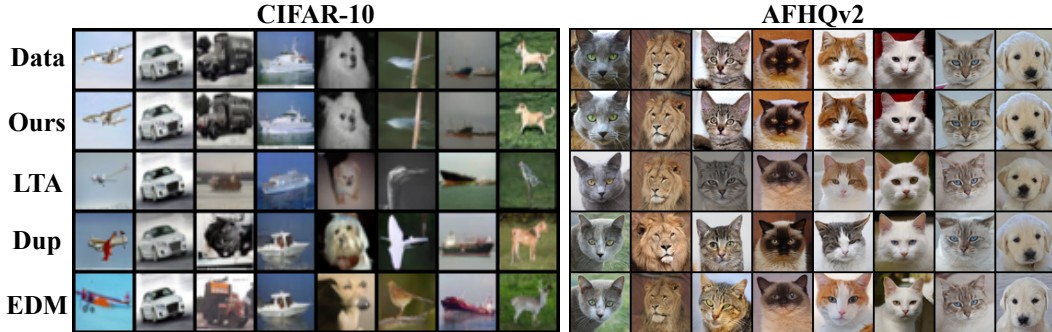

Figure 5: The uncurated samples of image exfiltration results of image diffusion models.

We present qualitative results for unconditional generation in Figure 5. For text-to-image generation results, please refer to Appendix G. The matched pairs follow our evaluation procedure, where replicated images from all methods have top-1 SSCD similarities greater than 0.5. To compare replication quality, we randomly sampled training images that were replicated by all methods. As shown in Figure 5, all methods except ours exhibit inconsistencies in texture, color temperature, and object orientation. Only our approach accurately reproduces the original training data. For FID comparisons, please see Appendix F.

## 6 CONCLUSION AND FUTURE WORK

In this paper, our research highlights a significant yet underexplored vulnerability within diffusion models, specifically their susceptibility to adversarial backdoor attacks that could potentially enable data exfiltration. By pioneering the use of trigger embeddings as a novel method for backdoor insertion, we not only expose a critical security flaw in generative AI systems but also provide a unique lens through which to examine their resilience against sophisticated cyber threats. Our findings concerning the implanting of backdoors into text-to-image diffusion models, including the innovative Caption Backdoor Subnet (CBS), underscore the urgent need for the development of advanced defensive strategies to safeguard these models from such vulnerabilities. As AI integrates across sectors, securing generative models is crucial. Our work establishes a foundation for research to protect AI against emerging threats, fostering more secure and reliable systems. In future work, we aim to examine how current defense mechanisms, such as those proposed in [2; 15], impact the efficacy of our introduced backdoor attack. This investigation will contribute to a deeper understanding of the robustness of existing defense strategies against our novel backdoor technique leveraging diffusion models.

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

## A  TRAINING CONFIGURATIONS

For all experiments, the ratio of training batch size for diffusion loss ($\mathcal{L}_t^{\mathrm{DM}}$) and backdoor loss ($\mathcal{L}_t^{\mathrm{Trig}}$ and $\mathcal{L}^{\mathrm{C}}$) is set to be $1:1$. For example, if the batch size is $512$, we divide it into $256$ and $256$ for diffusion loss and backdoor loss respectively. For unconditional image generation, the training configuration follows the same setup as EDM [17]. For text-to-image generation, we adhere to most of the configuration of SD, except the batch size is set to $16$ due to the computational resources limitation. For the CBS, the caption label threshold $\tau$ used to determine the presence of tokens is set to $0.8$, and the layer size threshold $n_w$ for defining the small layer is set to $10^4$.

## B  TRAINING DETAILS FOR DIFFUSION MODELS

We have demonstrated the effectiveness of our backdoor approach in unconditional image diffusion by adhering to the architecture and training loss of the EDM. To achieve this, we follow the default configuration with `--arch=ddpmpp` as provided in the official code of EDM [17], for both the CIFAR-10 and AFHQv2 dataset. We mute the flipping and augmentation for the trigger batch in CIFAR-10 dataset while preserving it for the normal batch to avoid influencing the performance of the backdoored model for unconditional generation. For both datasets, we set the training iteration to 100000k images iteration. We provide the training procedure for Diffusion Model in Algorithm 1.

---
**Algorithm 1** Diffusion Model Training Procedure
---
**Input: Dataset** $\mathcal{D}$, **Model** $\theta$, **Trigger Generating Function (TGF)** $\mathcal{F}$

1: **repeat**
2:     $\mathbf{x}_0, \hat{\mathbf{x}}_0 \sim \mathcal{D}$, $i :=$ indices of $\hat{\mathbf{x}}_0$ in $\mathcal{D}$
3:     $\mathbf{e}_u = \mathbf{0}^{d_t}$, $\hat{\mathbf{e}}_u = \mathcal{F}(i)$                    $\triangleright$ $\mathbf{e}_u$ is a zero vector with dimension $d_t$
4:     $\ddot{\mathbf{x}}_0 = [\mathbf{x}_0, \hat{\mathbf{x}}_0]$, $\ddot{\mathbf{e}}_u = [\mathbf{e}_u, \hat{\mathbf{e}}_u]$
5:     $t \sim \mathrm{Uniform}(1, \cdots, T)$, $\boldsymbol{\epsilon} \sim \mathcal{N}(\mathbf{0}, \mathbf{I})$, $\mathbf{e}_t :=$ embedding for $t$
6:     $\ddot{\mathbf{x}}_t = \sqrt{\alpha_t}\ddot{\mathbf{x}}_0 + \sqrt{1 - \alpha_t}\boldsymbol{\epsilon}$
7:     $\mathcal{L}_t^{\mathrm{DM}}(\boldsymbol{\theta}) + \gamma\mathcal{L}_t^{\mathrm{Trig}}(\boldsymbol{\theta}) = \left\| \boldsymbol{\epsilon} - \boldsymbol{\epsilon_\theta}(\ddot{\mathbf{x}}_t, \mathbf{e}_t + \ddot{\mathbf{e}}_{\mathbf{u}}) \right\|^2$
8:     Taking gradient step on $\nabla_\theta \mathcal{L}_t^{\mathrm{DM}}(\boldsymbol{\theta}) + \mathcal{L}_t^{\mathrm{Trig}}(\boldsymbol{\theta})$
9: **until** converged

---

## C  TRAINING DETAILS FOR TEXT-TO-IMAGE DIFFUSION MODELS

In order to inject a backdoor into Text-To-Image diffusion models that can be used for image and caption exfiltration, we first train a network to overfit the caption data explicitly. This network is then used to provide the pre-trained weights that will be used to initialize the CBS network. The CBS network is created by randomly selecting parameters from the U-Net layers. It is worth noting that it is possible to train the CBS network from scratch without using explicitly trained network weights. However, using a set of effective weights can speed up the model's convergence. We provide the training procedure for Text-To-Image Diffusion Model in Algorithm 2.

## D  DESIGN OF TGF

As mentioned, *Uniqueness*, *Similarity Consistency* and *Dimensionality* must be satisfied in the design of TGF. We construct a comparative table to detail the characteristics of various encoding functions and clarify the design of TGF. In Table 4, we list the properties of various encoding functions.

**One-Hot encoding** converts categorical values into binary vectors with only one high (1) value and the rest low (0), exemplified by encoding "Red", "Green", and "Blue" as [1, 0, 0], [0, 1, 0], and [0, 0, 1] respectively, The dimension of embedding depend on the size of category.

**Hash encoding** maps categorical values to fixed-size vectors using a hash function, for instance, encoding "Apple", "Banana", and "Cherry" into 3-bit vectors might result in "Apple" as [1, 0,

**Algorithm 2** Text-To-Image Diffusion Model Training Procedure
___
    **Input: Dataset** $\mathcal{D}$, **Model** $\theta$, **TGF** $\mathcal{F}, \mathcal{F}_c$, **Initialize weights for CBS** $\tilde{\mathcal{W}}$
1: $\mathcal{M}_\phi \leftarrow \{\mathbf{W}_1, \mathbf{W}_2, \cdots, \mathbf{W}_m \,|\, \mathbf{W}_m \subset \theta\}$                 $\triangleright$ Select $\phi$ from U-Net in $\theta$
2: Initialize $\mathcal{M}_\phi$ with $\tilde{\mathcal{W}}$
3: **repeat**
4:      $\mathbf{x}_0, \hat{\mathbf{x}}_0, C_p, p \sim \mathcal{D}, i := $ indices of $\hat{\mathbf{x}}_0$ in $\mathcal{D}, i_c := $ indices of $p$ in $\mathcal{D}$
5:      $\mathbf{e}_c = \mathcal{F}(i), \mathbf{e}_c^p = \mathcal{F}_c(i_c)$          $\triangleright$ $\mathbf{e}_c$ and $\mathbf{e}_c^p$ are trigger embeddings for $x_0$ and $p$
6:      $\ddot{\mathbf{x}}_0 = [\mathbf{x}_0, \hat{\mathbf{x}}_0], \ddot{\mathbf{e}}_u = [\mathbf{e}_p, \mathbf{e}_c]$                  $\triangleright$ $\mathbf{e}_p$ is text embedding of $p$
7:      $t \sim \text{Uniform}(1, \cdots, T), \boldsymbol{\epsilon} \sim \mathcal{N}(\mathbf{0}, \mathbf{I})$
8:      $\ddot{\mathbf{x}}_t = \sqrt{\alpha_t} \ddot{\mathbf{x}}_0 + \sqrt{1 - \alpha_t} \boldsymbol{\epsilon}$
9:      $\mathcal{L}_t^{\text{DM}}(\boldsymbol{\theta}) + \gamma \mathcal{L}_t^{\text{Trig}}(\boldsymbol{\theta}) = \left\| \boldsymbol{\epsilon} - \boldsymbol{\epsilon}_{\boldsymbol{\theta}}(\ddot{\mathbf{x}}_t, t, \ddot{e}_u) \right\|^2$
10:      $\mathcal{L}^{\text{C}}(\boldsymbol{\theta}) = \left\| \mathcal{M}_\phi(\mathbf{e}_c^p) - C_p \right\|^2$
11:      Taking gradient step on $\nabla_\theta \mathcal{L}_t^{\text{DM}}(\boldsymbol{\theta}) + \mathcal{L}_t^{\text{Trig}}(\boldsymbol{\theta}) + \mathcal{L}^{\text{C}}(\boldsymbol{\theta})$
12: **until** converged
___

Table 4: Properties of various encoding functions, where $d$ is the dimension of embedding.

| Encding Function | Uniqueness | Consistent Similarity | Flexible Dimension | Time Complexity |
|---|---|---|---|---|
| One-Hot | ✓ | ✓ | ✗ | $O(1)$ |
| Hash | ✗ | ✓ | ✓ | $O(1)$ |
| Binary | ✓ | ✗ | ✗ | $O(1)$ |
| Fourier | ✓ | ✗ | ✓ | $O(1)$ |
| DHE | ✓ | ✓ | ✓ | $O(d)$ |
| Uniform | ✓ | ✓ | ✓ | $O(1)$ |

0], "Banana" as [0, 1, 0], and "Cherry" as [1, 1, 0], depending on the hash function's distribution. Different inputs can result in the same output due to collisions.

**Binary encoding** represents integers in binary form, which does not meet similarity consistency property such as how $f(12) = [1, 1, 0, 0]$ is closer to $f(13) = [1, 1, 0, 1]$ than to $f(7) = [0, 0, 1, 1]$.

**Fourier Feature encoding** [42] transforms input features into a high-dimensional space using sinusoidal functions, enhancing a model's ability to learn high-frequency patterns. Mathematically, it's expressed as $z = [sin(2\pi Bx), cos(2\pi Bx)]$, where $z$ is the encoded feature vector, $x$ the input, and $B$ a matrix or vector of frequencies, improving the model's pattern recognition capabilities.

**Deep Hash Embedding (DHE)** [16] is an encoding function used in recommendation systems. DHE encodes feature values into unique identifier vectors using multiple hashing functions and transformations. Given the computational effort involved in multiple hashing, the time complexity of DHE is $O(d)$, where $d$ is the dimension of embedding. For more details, we refer the reader to the official paper.

# E   MODEL SELECTION FOR CBS NETWORK

The architecture of the CBS network may affect its effectiveness in learning the mapping between the trigger and the caption data. As mentioned in Section C in supplementary material, we train a network to overfit the caption data and use it to initialize the CBS network. We show the performance of this network in recovering the original caption across different architectures in Table 5. Since the model with $\mathbf{W}_1^{(128 \times 256)} \times \mathbf{W}_2^{(256 \times d_\mathcal{T})}$ achieves the best performance and has a moderate number of parameters, we select it as the architecture for the CBS network in our experiments.

Table 5: "The illustration demonstrates the performance of CBS-initialized weights across different model architectures. The upper section of the table depicts variations in the input dimension, specifically the dimension of the trigger embedding for the caption. The lower section of the table illustrates the changes in the number of layers within the model.

| Model | BLEU ↑ | BERT Score ↑ | ROUGE ↑ | | |
|---|---|---|---|---|---|
| | | | 1 | 2 | L |
| $\mathbf{W_1}^{(32 \times 256)} \times \mathbf{W_2}^{(256 \times d_\mathcal{T})}$ | 0.366 | 0.949 | 0.863 | 0.516 | 0.674 |
| $\mathbf{W_1}^{(128 \times 256)} \times \mathbf{W_2}^{(256 \times d_\mathcal{T})}$ | **0.412** | **0.949** | **0.876** | **0.546** | **0.682** |
| $\mathbf{W_1}^{(512 \times 256)} \times \mathbf{W_2}^{(256 \times d_\mathcal{T})}$ | 0.383 | 0.944 | 0.825 | 0.523 | 0.659 |
| $\mathbf{W_1}^{(128 \times d_\mathcal{T})}$ | *Model not converging* | | | | |
| $\mathbf{W_1}^{(128 \times 256)} \times \mathbf{W_2}^{(256 \times d_\mathcal{T})}$ | **0.412** | 0.949 | 0.876 | 0.546 | 0.682 |
| $\mathbf{W_1}^{(128 \times 256)} \times \mathbf{W_2}^{(256 \times 512)} \times \mathbf{W_3}^{(512 \times d_\mathcal{T})}$ | 0.407 | **0.954** | **0.883** | **0.555** | **0.693** |

Table 6: FID of generated benign images and triggered images on CIFAR-10 and AFHQv2 datasets. Note that EDM and EDM+Dup do not have an explicit trigger mechanism, so the triggered FID cannot be calculated. Moreover, since EDM+LTA is based on pretrained EDM, hence the benign FID scores are consistent with those of the original EDM.

| Method | CIFAR-10 ($32 \times 32$) | | AFHQv2 ($64 \times 64$) | |
|---|---|---|---|---|
| | Benign | Triggered | Benign | Triggered |
| EDM [17] | **2.00** | - | **2.11** | - |
| EDM + Dup [37] (N=15) | 2.76 | - | 3.58 | - |
| EDM + LTA [51] (M=200k) | **2.00** | 80.19 | **2.11** | 63.22 |
| EDM + TGF (ours) | 2.44 | **1.92** | 2.29 | **1.09** |

Figure 6: The uncurated samples of image exfiltration results of image diffusion models.

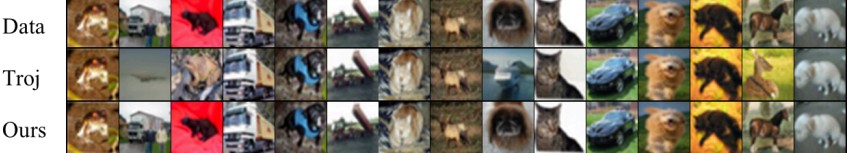

## F  FID OF IMAGE DIFFUSION MODELS

We demonstrate that integrating a backdoor approach does not compromise the image generation capabilities of a diffusion model. In Table 6, we present the FID scores of the backdoored EDM enhanced by our TGF, alongside various exfiltration approaches based on EDM. Our findings indicate that our backdoored model retains the generation capabilities of the original diffusion model, as evidenced by FID scores of 2.44 and 2.29 for CIFAR-10 and AFHQv2, respectively. However, duplicating training data leads to a degradation in FID scores, particularly on AFHQv2, where the FID score deteriorates from 2.11 to 3.58. For the loss threshold attack approach, although the benign FID is the same as the original, the triggered FID is drastically degraded due to the limited diversity of generated images.

## G  QUALITATIVE RESULTS

For qualitative results of unconditional generation compare to Trojdiff [5], text-conditional image generation are shown in Figure 6 and Figure 7 respectively. Which illustrates the differences between the recovered and original images in the dataset. Additionally, Figure 8 shows the examples of caption reconstruction.

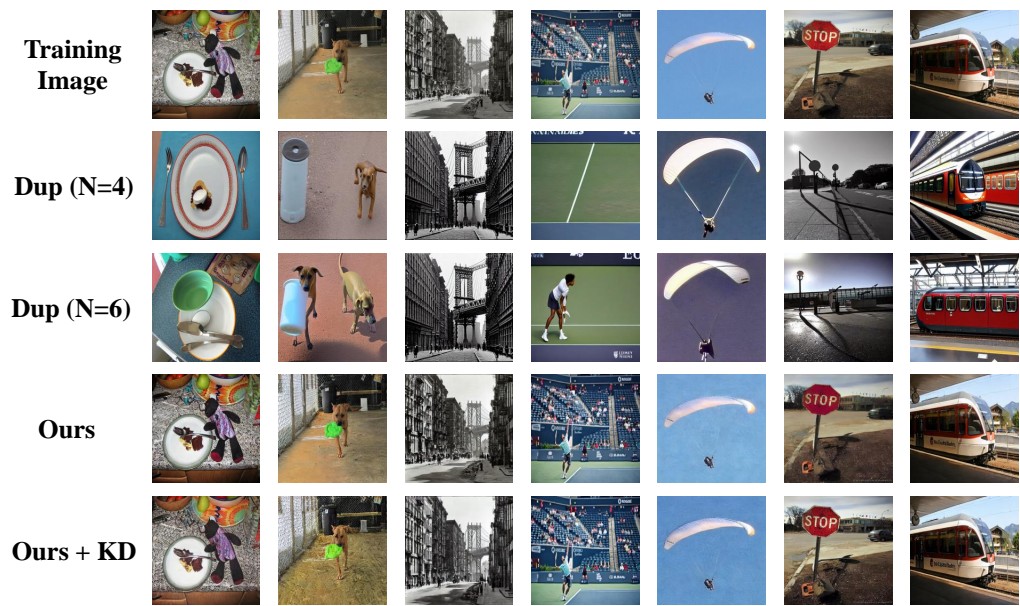

Figure 7: Qualitative Result of image exfiltration in Text-To-Image Diffusion Model.

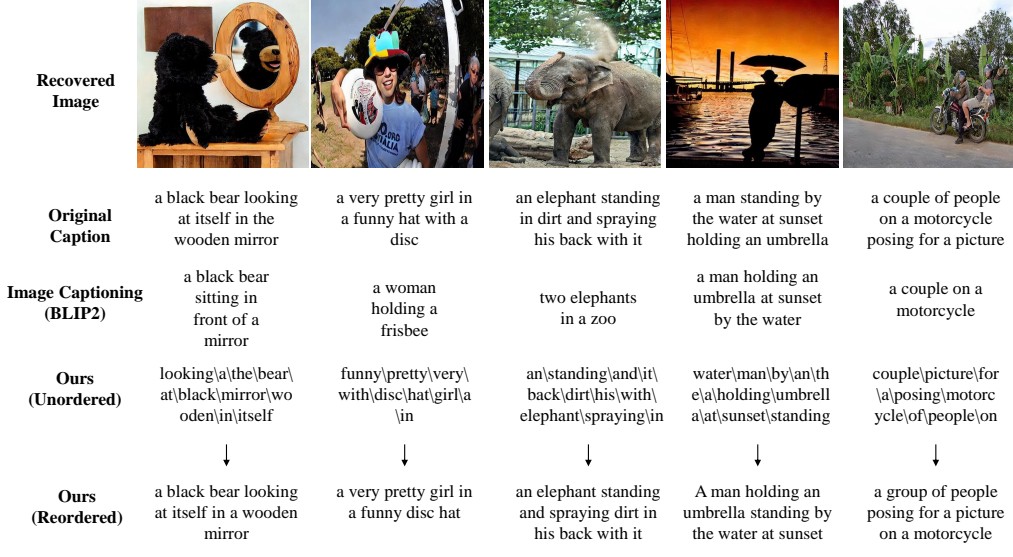

Figure 8: The figure presents a side-by-side comparison of image captions: those generated by an image captioning model versus those decoded and restructured using the Caption Backdoor Subnet (CBS) and reorder by a Language Model (LLM), showcasing the nuanced capabilities of the CBS network in caption recovery and organization.

## H  ABLATION STUDY ON DESIGN OF TGF

In this section, we examine the criteria for selecting an appropriate encoding method for the Trigger Generating Function (TGF), we construct a comparative table to detail the characteristics of various encoding functions, evaluating them across three aspects: *Uniqueness*, *Consistent Similarity*, and *Dimensionality*, in addition to assessing their computational efficiency in generating embeddings.

To validate our hypothesis, we select three encoding methods that meet these criteria (i.e. Uniform, Fourier [42] and DHE Encoding) for our trigger embeddings and conducted a backdoor training process on the CIFAR-10 dataset. The outcomes, presented in Table 7, reveal that models trained

Table 7: Assessment of unconditional image generation performance in benign and backdoor scenarios on CIFAR-10 datasets using varied encoding functions in Trigger Generating Function (TGF). All the experiments are based on EDM [17].

| TGF | Benign | Triggered | | | | | | | |
|---|---|---|---|---|---|---|---|---|---|
| | FID ↓ | SSIM ↑ | LPIPS ↓ | L2 ↓ | SSCD > 0.5 | | SSCD > 0.7 | |
| | | | | | Precision | Recall | Precision | Recall |
| Fourier Encoding | *Model not converging* | | | | | | | |
| DHE Encoding [16] | 4.31 | 0.582 | 0.236 | 0.128 | 0.969 | 0.880 | 0.277 | 0.273 |
| Uniform Encoding (ours) | 2.00 | 0.637 | 0.205 | 0.119 | 0.980 | 0.932 | 0.350 | 0.347 |

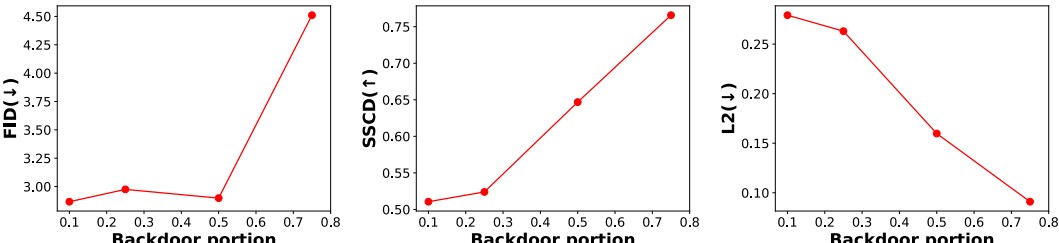

Figure 9: Comparison of backdoor and benign performance with varying backdoor portions. The experiment is conducted with 5000 trigger-target pairs in EDM.

using fourier features encoding TGF failed to converge. This issue is attributed to the collision of features between fourier features encoding and timestep encoding (i.e. positional encoding), both of which utilize sinusoidal functions. Conversely, we observe that both DHE and Uniform encoding are viable for TGF, effectively generating trigger embeddings that support our backdoor methodology. However, the computational demand for generating DHE embeddings is significantly high, making it a less efficient choice. Consequently, we opted for Uniform encoding as the preferred TGF encoding in our experiments.

# I ADDITIONAL EXPERIMENTS

In this section, we present additional experiments to demonstrate the effectiveness of our backdoored method.

## I.1 ABLATION ON BACKDOOR PORTION

In this experiment, conducted within the EDM framework with 5000 trigger-target pairs, we progressively increase the backdoor portion from 0 to 1. As shown in Figure 9, when the backdoor portion is set to 0.5, it achieves an optimal balance between the quality of reconstructed images and the diversity and high quality of generated images.

## I.2 RESULTS OF TEXT-TO-IMAGE ON LAION DATASET

We further extend our experiments to text-to-image diffusion models using the LAION subset dataset, comprising 11k training images, employing 500 trigger-target pairs. We compare this with the SD+Dup setting in Table 8, where trigger-target pairs are duplicated 6 times in the training dataset. The result aligns with our findings from the COCO dataset experiments presented in Table 3 of our main paper. Our method successfully recovers the training images, showing at least a 0.428 improvement in the SSCD metric.

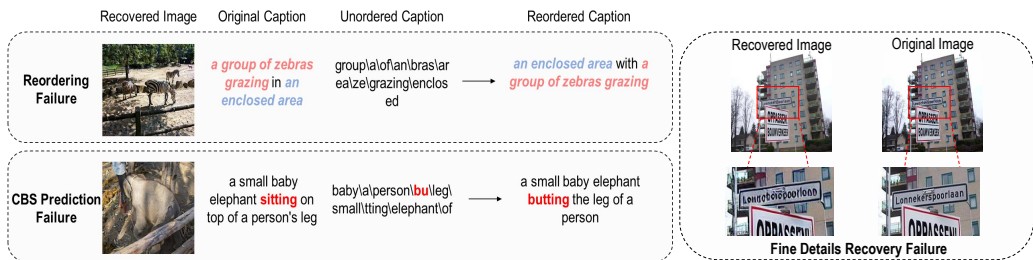

Figure 10: The illustration of three failure cases in our backdoor approach for data exfiltration: Reordering Failure, CBS prediction failure, and fine-details recovery failure.

Table 8: Comparative analysis of text-to-image diffusion models in pretrained and finetuned states using our backdoor settings for image exfiltration. Evaluation conducted on the LAION dataset with 500 trigger-target pairs.

| Method | Benign | | Triggered | | | |
|---|---|---|---|---|---|---|
| | CLIP Score↑ | IS↑ | L2↓ | SSIM↑ | LPIPS↓ | SSCD↑ |
| SD Pretrained | 29.7811 | 28.3302 ± 1.36 | - | - | - | - |
| SD + Dup (N=6) | 27.8679 | 21.6310 ± 0.70 | 0.1329 | 0.1414 | 0.7334 | 0.1359 |
| SD + TGF (ours) | 28.4637 | 25.7616 ± 1.51 | **0.0486** | **0.3722** | **0.4765** | **0.6518** |
| SD + TGF + KD (ours) | **29.9715** | **28.7779 ± 1.57** | 0.0603 | 0.3151 | 0.5287 | 0.5640 |

## J    LIMITATIONS

Our research demonstrates the feasibility of implanting a backdoor into diffusion models for data exfiltration. However, it faces limitations, which are highlighted in Figure 10. These include Reordering Failure, where LLMs may incorrectly invert the order of sentences; CBS Prediction Failure, which points to potential errors in CBS predictions; and Fine Details Recovery Failure, reflecting the model's struggles to accurately restore minor features like small textual elements within images. These challenges underline the need for further refinement of our method.

## K    ETHICAL CONSIDERATIONS

We recognize the importance of ethics in AI security research and are committed to expanding our discussion on potential implications and safeguards. Below, we include a more in-depth analysis of the ethical challenges posed by our method, along with a risk assessment, proposed countermeasures, and considerations for data ethics.

### K.1    POTENTIAL MISUSE

The proposed technique presents significant risks, particularly in the context of insider threats within secure environments. The potential misuse involves exploiting access to high-quality datasets during the training phase to insert latent backdoors into models. This could enable attackers to covertly exfiltrate sensitive customer information. Additionally, through sophisticated data conversion techniques, highly sensitive information—such as fingerprint data or bank account numbers—could be transformed into images and extracted alongside other data, further exacerbating the risk of data breaches and unauthorized disclosure.

### K.2    DEFENSE MECHANISM

To mitigate the risks associated with our proposed backdoor technique, we suggest two primary approaches.

Table 9: Performance of backdoor models on image exfiltration in uncondtional image diffusion post-defense with different portion of clean dataset. FT means fine-tuning here.

| Method | Dataset Ratio | Benign | Triggered | | | |
| --- | --- | --- | --- | --- | --- | --- |
| | | FID ↓ | L2 ↓ | SSIM ↑ | LPIPS↓ | SSCD↑ |
| DDPM + TGF (Before FT) | - | 5.1738 | 0.0131 | 0.9942 | 0.0060 | 0.9756 |
| DDPM + TGF (After FT) | 0.5 | 5.3019 | 0.1558 | 0.5124 | 0.3354 | 0.5792 |
| DDPM + TGF (After FT) | 1.0 | 4.8959 | 0.1955 | 0.3203 | 0.4665 | 0.4591 |

Table 10: Performance of backdoor models on image exfiltration in text-to-image diffusion post-defense, with red numbers showing changes.

| Method | Benign | | Triggered | | | |
| --- | --- | --- | --- | --- | --- | --- |
| | CLIP Score↑ | IS↑ | L2 ↓ | SSIM ↑ | LPIPS ↓ | SSCD↑ |
| SD + Dup [37] (N=6) | 28.103 | 29.65 ± 0.96 | 0.167 (↑ 0.02) | 0.123 (↓ 0.03) | 0.774 (↑ 0.04) | 0.062 (↓ 0.06) |
| SD + TGF (ours) | 28.660 | 33.19 ± 0.93 | **0.029** (↑ 0.02) | **0.546** (↓ 0.21) | **0.381** (↑ 0.15) | **0.689** (↓ 0.21) |
| SD + TGF + KD (ours) | **28.846** | **33.33 ± 0.60** | 0.034 (↑ 0.02) | 0.510 (↓ 0.17) | 0.404 (↑ 0.13) | 0.657 (↓ 0.19) |

First, **Early Detection Before Model Release**: Although performance degradation may not be directly observable in compromised models, certain indicators can signal the presence of a backdoor. Specifically, the training or fine-tuning duration tends to be longer, and the convergence speed slower, compared to unaffected models. A rigorous review of training resources prior to model release could help detect potential backdoor injections.

Second, **Model Recovery from Backdoors**: We recommend implementing a fine-tuning strategy using clean samples. Previous research [33; 56] has demonstrated the effectiveness of this approach in neutralizing backdoors in machine learning models. Specifically, we propose fine-tuning the suspect model with a carefully curated portion of clean, uncontaminated data. To validate the effectiveness of this method in eliminating backdoors, we conducted an experiment, focusing on both backdoor and benign performance before and after fine-tuning with clean samples.

As shown in Table 9 (for the unconditional image diffusion model), Table 10 (for the text-conditioned diffusion model), and Table 11 (for caption extraction), our method exhibited significant performance changes post-fine-tuning. The quality of the reconstructed images degraded substantially, while benign performance remained relatively stable. This demonstrates that fine-tuning with clean samples can effectively mitigate the effects of backdoors.

### K.3 TRAINING DATA ETHICS

In our study, we prioritize the ethical selection and processing of datasets. For image diffusion models, we use the complete datasets of CIFAR-10, AFHQv2, and ImageNet. For text-to-image diffusion models, we work with a curated subset of 3,000 images from MS-COCO and 11,000 images from LAION.

To ensure the ethical use of these datasets, we implement several robust measures. We apply strict content filters to eliminate potentially sensitive or problematic images using the safety checker pretrained by the CompVis community. Specifically, as detailed in [27], we calculate the cosine similarity between the image embeddings and 17 fixed embedding vectors representing sensitive

Table 11: Performance of backdoor models on caption exfiltration in text-to-image diffusion post-defense, as settings in Table 10.

| Method | BLEU ↑ | BERT Score ↑ | ROUGE ↑ | | |
| --- | --- | --- | --- | --- | --- |
| | | | 1 | 2 | L |
| CBS (ours) | **0.385** (↓ 0.006) | **0.950** (↓ 0.001) | 0.862 (↓ 0.013) | **0.522** (↓ 0.010) | **0.673** (↓ 0.009) |
| CBS + KD (ours) | 0.359 (↓ 0.043) | 0.944 (↓ 0.005) | **0.863** (↓ 0.014) | 0.504 (↓ 0.046) | 0.662 (↓ 0.021) |

concepts. If the similarity exceeds a predefined threshold, the image is flagged as problematic and subsequently removed.

These measures are crucial for maintaining research integrity and addressing ethical data usage. We recognize the challenges and are committed to refining our data ethics practices.

