# OpenReview forum: "Data Exfiltration in Diffusion Models: A Backdoor Attack Approach"
_ICLR.cc/2025/Conference — ICLR 2025 Conference Withdrawn Submission_

### Official Review · Reviewer_srex · 2024-10-30

**Soundness:** 3
**Presentation:** 4
**Contribution:** 3
**Rating:** 6
**Confidence:** 4

**Summary:**

This paper delves into a novel approach for data exfiltration by strategically implanting backdoors into diffusion models. Data exfiltration through backdoor attacks is a formidable task as it demands that the model memorize the entire dataset without compromising the normal diversity of images. However, memorization and diversification often conflict with each other. To address this issue, the paper presents unique trigger embeddings for image exfiltration. Additionally, to overcome the challenge of extracting corresponding captions, a Caption Backdoor Subnet (CBS) module is trained. This paper resolves an important security-related question through a reasonable method.

**Strengths:**

1. The problem addressed is extremely crucial and the application scenario is rather novel.
2. The design of the method and the model structure are relatively reasonable.
3. The writing and organization of this paper is very clean and well-followed.

**Weaknesses:**

1. **Too strong attacker’s capabilities assumption**. The attacker is depicted as being extremely capable of attacking, having access to all training data, mastering the training and inference processes of the model, and obtaining the trained backdoored model. This raises doubts about whether such an attack scenario is practical in the real world.

2. **Lack of theoretic analysis**. How does $\mathbb{L}^{Trig}(\theta)$ and $\\mathcal{L}^C (\theta)$ impact the convergence of training? Why can we select the parameters from the U-Net layers as the parameters of CBS?

3. **More experiments should be conducted.** For example, 1) How do different TGFs impact the ASR of the backdoored model and the process of the training. 2) What’s the effectiveness of the model when applied to more complex data scenarios, such as the COCO datasets, rather than CIFAR10 and AFHQv2 datasets which are relatively simple with relatively less image details. 3) Since models containing sensitive data are generally subject to rigorous censorship, it is suggested that the authors add experiments to verify the effect of the model under existed trigger detection methods.

**Questions:**

see the weekness

---

> ### Author Response · Authors · 2024-11-23
>
> We greatly appreciate the reviewer’s detailed feedback on our work, highlighting the need for theoretical analysis and additional experiments, which helped identify areas to strengthen our manuscript.
>
> - **Reply to Q1** (Strong attacker assumption):
>
>     The practicality of our proposed backdoor attack depends on the assumed threat model, particularly the requirement of full access to the victim model during both the backdoor injection and triggering phases. However, as highlighted in Footnote 1, real-life incidents, such as the 2016 case involving Anthony Levandowski at Google and the 2022 incident involving Qian Sang at Yahoo, underscore the possibility of significant data security breaches. Please note that our setting aligns with the settings commonly studied in research on data exfiltration [1, 2] and backdoor attacks targeting diffusion models [3, 4, 5]. We believe that these assumptions mirror realistic scenarios in the real world.
>
>
> - **Reply to Q2** (Lack of theoretic analysis):
>
>     Thank you for your detailed feedback regarding the theoretical analysis of our proposed method and the impact of the loss terms on training convergence. Below, we address your concerns point by point:
>
>     1. Impact of $\mathcal{L_t^{\text{Trig}}}(\boldsymbol\theta)$:
>
>         In our framework, the original diffusion loss $\mathcal{L_t^{\text{DM}}}(\boldsymbol\theta)$ is designed to minimize the evidence lower bound (ELBO) between the real and generative distributions. Specifically, for text-to-image diffusion models, $\mathcal{L}_t^{\text{DM}}(\boldsymbol\theta)$ minimizes the following:
>
>         $$\lambda_t\mathcal{L_t^{\text{DM}}}(\boldsymbol\theta)=\mathbb{D_{\text{KL}}}\Big[q(x_{t-1}|x_t,x_0,c)\big\Vert p_{\boldsymbol\theta}(x_{t-1}|x_t,c,\varnothing)\Big],$$
>
>         where $c$ represents the text prompt and $\varnothing$ corresponds to the untriggered condition.
>
>         To introduce the backdoor attack, we modify the diffusion objective by incorporating $\mathcal{L_t^{\text{Trig}}}(\boldsymbol\theta,\mathbf{e_u^i})$, which uses a trigger embedding $\mathbf{e_u^i}$ as a condition to generate $x^i$. This adjustment minimizes the following:
>
>         $$\lambda_t\mathcal{L_t^{\text{Trig}}}(\boldsymbol\theta,\mathbf{e_u^i})=\mathbb{D_{\text{KL}}}\Big[q_{i}(x_{t-1}|x_t,x_0,c)\big\Vert p_{\boldsymbol\theta}(x_{t-1}|x_t,c,\mathbf{e_u^i})\Big],$$
>
>         where $i$ denotes the index of the training data and $q_{i}(x_0|c)=\delta(x_0-x^i)$ is a Dirac delta function representing the $i$-th data.
>
>         By minimizing both $\mathcal{L_t^{\text{DM}}}(\boldsymbol\theta)$ and $\mathcal{L_t^{\text{Trig}}}(\boldsymbol\theta,\mathbf{e}_u^i)$, we can:
>         - Generate normal data from $p_{\boldsymbol\theta}(x_0|c,\varnothing)$, and
>         - Extract training data by sampling $x^i\sim p_{\boldsymbol\theta}(x_0|c,\mathbf{e}_u^i)$.
>
>     2. Impact of $\mathcal{L^\text{C}}(\boldsymbol\theta)$ and CBS on the Diffusion Model:
>
>         We primarily design the loss based on the lottery ticket hypothesis [6]. Importantly, this loss term only affects a small subset of parameters within the U-Net architecture, and we limit the trainable weights to those sampled from sufficiently large layers. This localized influence of $\mathcal{L^\text{C}}(\boldsymbol\theta)$ ensures that the benign performance of the model remains stable. Additionally, the weights of the caption backdoor subnet (CBS) are trained jointly with the diffusion losses $\mathcal{L_t^{\text{DM}}}(\boldsymbol\theta)+\mathcal{L_t^{\text{Trig}}}(\boldsymbol\theta,\mathbf{e}_u^i)$. By jointly training the CBS and the diffusion model, shared parameters are preserved within the diffusion model, maintaining the validity of the lottery ticket hypothesis and minimizing the CBS's impact on the model’s overall capacity.
>
>     3. Empirical results on convergence and performance:
>
>         Empirical results confirm that incorporating $\mathcal{L_t^{\text{Trig}}}(\boldsymbol{\theta})$ and $\mathcal{L^\text{C}}(\boldsymbol{\theta})$ does not hinder model convergence. As demonstrated in Table 2 of the main paper (for text-to-image diffusion models) and Table 6 in Appendix F (for image diffusion models), experiments conducted across a range of datasets and model architectures consistently show stable convergence, even with the inclusion of these additional loss terms. Precisely, while the backdoor loss slightly reduces benign metrics like FID and CLIP scores, the impact is minimal and practically insignificant.
>
> [1] Stealing Your Data from Compressed Machine Learning Models, ACM DAC’20.
>
> [2] Transpose Attack: Stealing Datasets with Bidirectional Training, NDSS Symposium’24.
>
> [3] "How to backdoor diffusion models?", CVPR‘23.
>
> [4] Trojdiff: Trojan attacks on diffusion models with diverse targets, CVPR’23.
>
> [5] VillanDiffusion: A Unified Backdoor Attack Framework for Diffusion Models, NeurIPS’23.
>
> [6] Finding sparse, trainable neural networks, ICLR'19.

---

> ### Author Response · Authors · 2024-11-23
>
> - **Reply to Q3** (More experiments should be conducted):
>
>     Below, we provide experiments for the examples you have mentioned:
>
>     1. Impact of different TGFs on model performance
>
>         In our paper, Footnote 2 refers to Appendix H for a more detailed comparison of TGF. Thank you for pointing this out—we will clarify the comparison about different TGF in the main text. In Table 7 of Appendix H, we present a comparison of different functions used as TGF, including Fourier Encoding, DHE Encoding, and Uniform Encoding. The results suggest that Uniform Encoding generates trigger embeddings that effectively support our backdoor methodology in both benign and backdoored scenarios, with better performance compared to the other two encoding functions. Specifically, Fourier Encoding encounters convergence issues, while DHE Encoding shows relatively weaker performance. Additionally, Uniform Encoding requires the least computational resources, making the backdoor implantation process more efficient. These observations align with the hypothesis in Section 4.2 of the main paper, which proposes that an effective encoding function should satisfy the following three criteria: (1) Uniqueness, (2) Consistent Similarities, and (3) Dimensionality. Uniform Encoding, when applied within the range $[-k, k]$ for some $k \in \mathbb{R}^+$, appears to meet these requirements.
>
>     2. Complex data scenarios
>
>         | Method         | Benign FID $\downarrow$ | Triggered L2 $\downarrow$ | Triggered SSIM $\uparrow$ | Triggered LPIPS $\downarrow$ | Triggered SSCD $\uparrow$ |
>         |----------------|:----:|:----:|:----:|:----:|:----:|
>         |EDM             |4.1720|  -   |  -   |  -   |  -   |
>         |EDM + LTA       |4.1720|0.3089|0.0885|0.6699|0.2026|
>         |EDM + TGF (ours)|4.4025|0.0736|0.8059|0.1094|0.8395|
>
>         In response, we conduct additional experiments using a more complex dataset to address this concern. Specifically, we utilize ImageNet at 64x64 resolution to train our backdoor approach (EDM+TGF) and compare it with the Loss Threshold Attack (LTA).
>
>         To implement our backdoor, we fine-tune the pretrained EDM model using 500 trigger-target pairs. Table above illustrates that our backdoor method maintains its effectiveness even when applied to more complex training images. Notably, our approach demonstrates a 314%  improvement in SSCD compared to the LTA method.
>
>     3. Effectiveness against current trigger detection methods
>
>
>         In our proposed method for backdoor attacks in diffusion models, the trigger injection mechanism differs significantly from existing approaches [1,2,3,4,5]. Specifically, in image diffusion models, we embed the trigger into the timestep embedding, instead of following current methods [1,2,3], which inject the trigger into the noisy latent space to shift the generation distribution based on a trigger pattern. For text-to-image diffusion models, our approach replaces the caption embedding with a trigger embedding generated by our proposed Trigger Generation Framework (TGF). This facilitates the model to memorize the mapping between the trigger and the corresponding training image. In contrast, existing methods [4,5] typically use a single word or character as their trigger.
>
>         Due to these differences, the existing trigger detection techniques [6,7,8,9], which focus on identifying triggers by reversing the distribution shift in image diffusion models [6,7] or detecting trigger word in text-to-image diffusion models [8,9] are unlikely to be effective against our method. Consequently, our approach remains more challenging to detect with current detection techniques, further demonstrating its stealthiness and practical applicability.
>
>
> [1] "How to backdoor diffusion models?", CVPR‘23.
>
> [2] Trojdiff: Trojan attacks on diffusion models with diverse targets, CVPR’23.
>
> [3] VillanDiffusion: A Unified Backdoor Attack Framework for Diffusion Models, NeurIPS’23.
>
> [4] Rickrolling the Artist: Injecting Backdoors into Text Encoders for Text-to-Image Synthesis, ICCV’23.
>
> [5] Personalization as a shortcut for few-shot backdoor attack against text-to-image diffusion models, AAAI’24.
>
> [6] Elijah: Eliminating Backdoors Injected in Diffusion Models via Distribution Shift, AAAI’24.
>
> [7] TERD: A Unified Framework for Safeguarding Diffusion Models Against Backdoors, ICML’24.
>
> [8] T2IShield: Defending Against Backdoors on Text-to-Image Diffusion Models, ECCV’24.
>
> [9] Defending Text-to-image Diffusion Models: Surprising Efficacy of Textual Perturbations Against Backdoor Attacks, ECCV’24 Workshop.

---

### Official Review · Reviewer_FWZP · 2024-11-03

**Soundness:** 3
**Presentation:** 3
**Contribution:** 2
**Rating:** 6
**Confidence:** 5

**Summary:**

The paper proposed a novel idea to exploit the backdoor attack to review the secret of the training data. For the DDPM model, by exploiting the special time embedding as the trigger, the attacker who injected the backdoor can recover the training data that he wants to leak. For the T2I models, the author uses special text embeddings instead of time embeddings as the trigger to help retrieve the training data. A Caption Backdoor Subnet, whose weights are retrieved from the U-Net of the victim T2I model was proposed to help recover the caption of the training samples.

**Strengths:**

1. Good writing, easy to follow;

2. Novel idea and method, I have learned from, and been greatly inspired by the paper;

3. Extensive experiments and evaluations, demonstrating the proposed methods are really effective.

**Weaknesses:**

I love the idea of exploiting the backdoor attack to cause secret leakage, yet mainly concerned with the threat model of the paper before I could convince myself to vote for the acceptance of the paper.

Firstly, the proposed method requires the attacker to have full access to the victim model in both the backdoor injection phase and the triggering phase, for the special embeddings need to be added to the model. This could greatly impact the applicability of this method-- seems that only open-sourced model are vulnerable to the proposed attack.

Secondly, the motivation for the attack is unclear to me. I doubt whether an insider, as mentioned in Sec.3, would carry out such an attack when he seems capable of releasing the training data directly afterward.  The insider can also just copy the data secretly before they leave the organization. I suppose the scenario can be described in another way, where the insider wants to harm the interests of the company by jeopardizing the published model to be less privacy-preserving for the purpose of getting back at their employer. I suggest the author further discuss why the attack is practical by giving detailed examples as above. This could help highlight the threat of this newly disclosed attack.

**Questions:**

Seen in weakness.

---

> ### Author Response · Authors · 2024-11-20
>
> We would like to thank you for the insightful comments for our work. Below, we address the concerns raised in the feedback.
>
> 1. Concerns about the Threat Model and Applicability
>
>     Our proposed method indeed requires full access to the victim model during both the backdoor injection and triggering phases, which is a relatively strong assumption as compared with other threat models, such as data poisoning alone or model theft. However, as highlighted in Footnote 1, real-life incidents, such as the 2016 case involving Anthony Levandowski at Google and the 2022 incident involving Qian Sang at Yahoo, underscore the prevalence of significant data security breaches. Moreover, such environments may be widely prevalent. For instance, publicly available open-source diffusion models for image generation, such as DeepFloyd IF, DeciDiffusion, and PixArt-α, can be freely downloaded and fine-tuned by anyone, including non-expert users, who may inadvertently introduce backdoors by directly running the compromised or unsafe code.
>
>     It is worth noting that our attacker assumptions of the access to the training dataset and control over the training and inference processes, are consistent with existing research on backdoor attacks in diffusion models [1, 2, 3] and data exfiltration [4, 5].
>
> 2. Motivation for the Attack
>
>     Regarding your concern about the practicality of the attack, particularly why an insider might choose to execute a backdoor attack rather than directly downloading the training data, we would like to clarify the following. In many environments, direct access to training data may be restricted due to organizational policies or technical constraints, such as data being transmitted remotely or stored in secure, controlled-access locations. These limitations make it difficult for an insider to obtain the data directly. Moreover, downloading training data often leaves detectable traces in system logs, which could raise alarms and expose the insider's actions. In contrast, our proposed backdoor attack provides a more covert method of data extraction, significantly reducing the likelihood of detection. This makes it an appealing option for insiders who prioritize stealth and wish to evade monitoring systems.
>
>     We also agree with the scenario you proposed, where an insider aims to harm the company's interests by compromising the model's privacy-preserving nature. This perspective complements the motivations described in our paper. We will expand the revised version to explicitly highlight that the attack can also serve as a means of causing reputational damage, further enriching the threat model and its implications.
>
> ---
>
> We hope that this additional context helps clarify the broader applicability and motivations behind the attack. We strongly believe that the proposed backdoor attack addresses an important and emerging threat in deep learning and privacy-preserving systems, where model integrity is increasingly at risk.
>
> [1] VillanDiffusion: A Unified Backdoor Attack Framework for Diffusion Models, NeurIPS’23.
>
> [2] How to backdoor diffusion models?, CVPR’23.
>
> [3] Trojdiff: Trojan attacks on diffusion models with diverse targets, CVPR’23.
>
> [4] Stealing Your Data from Compressed Machine Learning Models, ACM DAC’20.
>
> [5] Transpose Attack: Stealing Datasets with Bidirectional Training, NDSS Symposium’24.

---

> > ### Comment · Reviewer_FWZP · 2024-11-25
> >
> > Thank you for your response, which has addressed most of my concerns. I'll thereby raise the overall rating to 6.
> >
> > Here are still some suggestions. 1. I'm still not very convinced by the claim that says 'downloading training data often leaves detectable traces in system logs, which could raise alarms and expose the insider's actions'. I believe manipulating the training process will also leave such traceable clues for the inspectors. 2. You may clarify your motivation by proposing more specific examples.
> >
> > Additionally, I agree with the author that the work "Extracting training data from diffusion models" by Carlini et al may not constitute a proper baseline, as it focuses on the natural capability of the diffusion model to unintentionally remember some of the training samples. However, I also agree with Reviewer Z4kF that a comparison to the simple prompting might be essential, as it can demonstrate that the proposed method can enhance privacy leakage, which may help highlight your contributions. While the comparison to other backdoor techniques in the paper shows the technical contribution of this work. It's better for both of them to be presented as baselines.

---

> > > ### Author Response · Authors · 2024-12-01
> > >
> > > We sincerely appreciate the reviewer’s thoughtful feedback and decision in raising the score for our paper. Below, we provide further elaboration to address your concerns:
> > >
> > >
> > > - **Reviewer Comment**: manipulating the training process will also leave such traceable clues for the inspectors.
> > >
> > >   **Response**:
> > >
> > >   We acknowledge the reviewer's point that manipulating the training process could indeed leave traceable clues for inspectors. However, detecting such manipulations is not straightforward due to several key reasons:
> > >
> > >
> > >   1. **Subtle Changes in Training Code**: The modifications made during the training process to implant a backdoor are intentionally subtle and designed to blend seamlessly with normal operations. For example, our payload code for backdoor implantation involves simply adding the trigger embedding generated by TGF into the trigger batch alongside the normal batch during the training process. This approach avoids introducing any highly suspicious code (such as fixed trigger words) that could be easily detected by static analysis tools for code security like Snyk Code or Semgrep.
> > >
> > >   2. **Bypassing Static Analysis Tools**: When our code is potentially detectable by existing static analysis tools, we argue that it is feasible to leverage Large Language Models (LLMs) to evade vulnerability detection, as demonstrated in previous research [1,2]. For instance, CodeBreaker [2] have shown that models such as GPT-4 can reframe suspicious code to bypass both traditional and LLM-based detection methods, while retaining the payload's core vulnerable functionality.
> > >
> > >
> > >   [1] TrojanPuzzle: Covertly Poisoning Code-Suggestion Models, S&P 24.
> > >
> > >   [2] An LLM-Assisted Easy-to-Trigger Backdoor Attack on Code Completion Models: Injecting Disguised Vulnerabilities against Strong Detection, USENIX’24.
> > >
> > > - **Reviewer Comment**: comparison to the simple prompting might be essential.
> > >
> > >   **Response**:
> > >   We appreciate the reviewer's suggestion regarding a comparison to simple prompting. While simple prompting may serve as a baseline for extracting images similar to the training data, it often requires significant computational effort. In future work, We aim to include more competitive and appropriate baselines in subsequent studies to ensure a fair comparison and achieve comparable performance.

---

### Official Review · Reviewer_Z4kF · 2024-11-04

**Soundness:** 2
**Presentation:** 1
**Contribution:** 2
**Rating:** 3
**Confidence:** 5

**Summary:**

The paper describes an attack that injects a backdoor into a diffusion model in order to extract both the image and text caption from the training data. The paper proceeds with describing a strong threat model and outlines a mechanism to train the diffusion model with multiple losses that both embed the attack and permit high utility of the resulting model. The results demonstrate high efficacy of the method compared to the baselines.

**Strengths:**

- an attack targets both extraction of images and text which is novel.
- the experiments include two datasets for extraction
- both exfiltration methods seem to be interesting as they leverage diffusion model capabilities, e.g. attack text embeddings instead of time embeddings.

**Weaknesses:**

- the paper is hard to follow and the concepts are not presented clearly, I think more work should be done to polish the text. e.g. this is a poorly written objective: ''Our goal is to develop an innovative approach...''
- Motivation and threat model: the adversary needs access to data, training and inference in order to make the attack happen, this sounds like an extremely strong threat model where the goal is to extract the data that the attacker has already access to. Provided examples of real-world relevance also permit simply downloading data.
- Related work: "Extracting training data from diffusion models" by Carlini et al, has demonstrated an easy way to extract data without any poisoning, comparison with this paper would help a lot. The authors focus on comparing with other backdoor attacks, but I think the baseline should be just prompting.

**Questions:**

address Threat model, related work and improve paper presentation.

---

> ### Author Response · Authors · 2024-11-20
>
> We appreciate the opportunity to clarify our contributions and address specific concerns. Below, we provide detailed responses to each point raised.
>
> - **Reply to Q1** (Paper presentation):
>
>     We respectfully disagree with the reviewer’s assessment that the paper presentation is poor. Based on feedback from the other three reviewers, the paper’s clarity and presentation received ratings of at least 3, which suggests that the contributions, motivations, and proposed solutions are effectively communicated.
>
>     However, we appreciate the specific example provided. The sentence, *"Our goal is to develop an innovative approach..."* can be improved for clarity. A more precise phrasing would be:
>
>     *"Our goal is to implant a backdoor in diffusion models, allowing for the covert extraction of their training data without compromising their benign generative capabilities."*
>
>     We will revise the text accordingly to enhance readability and precision. Thank you for pointing this out, as it helps us further refine our work.
>
>
> - **Reply to Q2** (Practical usage of data exfiltration / Strong attacker assumption):
>
>     Our attack scenario aligns with existing research on backdoor attacks in diffusion models, e.g., [1,2,3], and data exfiltration, e.g., [4,5], where the attacker requires access to the training dataset and control over the training and inference pipelines. While the adversary indeed requires initial access to the training data, our proposed method addresses a distinct security concern that differs significantly from direct data downloading. The key advantage of our approach lies in its covert nature - while direct data downloads are easily detectable through network monitoring, access logs, and security audits, our backdoor-based exfiltration method creates a subtle, persistent channel that can bypass conventional security measures and monitoring systems. This is particularly relevant in real-world scenarios such as cloud service providers handling sensitive client data, or outsourced ML development, where direct downloads would trigger immediate security alerts. Our method effectively embeds a backdoor into the model while maintaining its original functionality. This makes the backdoor much harder to detect using standard security protocols, emphasizing its significance as a critical threat that demands careful attention in machine learning security.
>
>
> [1] VillanDiffusion: A Unified Backdoor Attack Framework for Diffusion Models, NeurIPS’23.
>
> [2] How to backdoor diffusion models?, CVPR’23.
>
> [3] Trojdiff: Trojan attacks on diffusion models with diverse targets, CVPR’23.
>
> [4] Stealing Your Data from Compressed Machine Learning Models, ACM DAC’20.
>
> [5] Transpose Attack: Stealing Datasets with Bidirectional Training, NDSS Symposium’24.
>
>
> - **Reply to Q3** (Weak baseline):
>
>     We would like to clarify that *"Extracting Training Data from Diffusion Models"* primarily investigates whether diffusion models memorize training data and can unintentionally reproduce it during inference. However, their method is impractical as an attack due to its intensive computational requirements. Specifically, their paper examines two categories of diffusion models:
>
>     1. **Text-to-Image Diffusion Models** (Section 4.2):
>
>         Their approach involves a two-step process: generating a massive number of images and applying membership inference techniques to filter candidates resembling the training data. However, this method is highly inefficient. For instance, in their experiments with Stable Diffusion, they generated *175 million* images but identified only *94* images as highly similar to the training data, based on a threshold of 0.15 under the L2 distance function.
>
>     2. **Unconditional Diffusion Models** (Sections 5.1 and 5.2):
>
>         The authors evaluate untargeted attacks to confirm that smaller models also exhibit memorization. They leverage membership inference techniques, including white-box attacks like the Loss Threshold Attack (LTA). Notably, we have already included a comparison with LTA in Table 1 of our paper.
>
>     While the work by Carlini et al. provides valuable insights into memorization within diffusion models, its objectives and methodology differ fundamentally from our focus on backdoor attacks. We appreciate your suggestion and will revise the related work section to more clearly delineate these distinctions and enhance the comprehensiveness of the discussion.

---

> > ### Comment · Reviewer_Z4kF · 2024-11-24
> >
> > thank you for the response, however the response did not address my concerns on the practicality of the attack nor the comparison to the provided paper:
> >
> > ` while direct data downloads are easily detectable through network monitoring, access logs, and security audits, our backdoor-based exfiltration method creates a subtle, persistent channel that can bypass conventional security measures and monitoring systems. ` -- this claim is unsupported, I would recommend providing more evidence that data download is easily detectable unlike backdoor attack --> the model owner can similarly make sure all generated images are checked against training dataset and ban them.
> >
> > `related work` the result with 175 million is only relevant when the attacker does not know what to generate, in the case of your attacker has access to the training data so it won't be difficult to extract the data given the known captions or image content.
> >
> > overall, I will keep my score

---

> > > ### Author Response · Authors · 2024-11-26
> > >
> > > We sincerely thank the reviewer for detailed feedback and for highlighting important aspects of our work. Below, we provide further elaboration to address your concerns:
> > >
> > > - **Reviewer Comment**:  "This claim is unsupported, I would recommend providing more evidence that data download is easily detectable unlike backdoor attack."
> > >
> > >   **Response**:
> > >
> > >    We’re surprised by this comment since this distinction is precisely why backdoor attacks are a significant concern in security research. Unlike unauthorized data downloads, which often leave detectable traces such as unusual network activity or access logs, backdoor attacks are specifically designed to evade detection by embedding malicious triggers that activate under very specific conditions.
> > >
> > >
> > >   To clarify:
> > >
> > >   1. **Detectability of Data Download**: Unauthorized data downloads typically result in detectable patterns, such as large volumes of data transfer, atypical access requests, or deviations from expected usage patterns. These anomalies are well-documented and monitored by intrusion detection systems and network security tools [1,2].
> > >
> > >   2. **Challenge of Backdoor Attacks**: Backdoor attacks, by contrast, are inherently more difficult to detect because they remain dormant under normal operations. They are activated only under specific conditions, often crafted to mimic legitimate behavior, making them invisible to standard monitoring systems.
> > >
> > >   To strengthen our argument, we have added references to relevant literature [3,4,5] that highlight these distinctions. We have also revised the corresponding section of the paper to provide a clearer explanation and additional evidence.
> > >
> > >
> > > - **Reviewer Comment**: “ the result with 175 million is only relevant when the attacker does not know what to generate”
> > >
> > >
> > >   **Response**:
> > >
> > >   Regarding your observation that the result involving 175 million images is only relevant when the attacker does not know what to generate, we would like to clarify this point. In the referenced study [6], the authors explicitly utilized the training data to guide their evaluation, meaning that the 175 million images were generated under the assumption that the attacker knows the target to generate. Specifically, they leveraged prompts derived from duplicated training data to generate images. This targeted approach aimed to improve the efficiency of verifying whether the diffusion model memorized specific data. Moreover, we note that your comment assumes the attacker has access to both the model and the training data during the exfiltration phase. However, this does not align with the threat model outlined in our paper. In our proposed attack framework, the attacker has simultaneous access to the model and training data only during the implantation phase, where the backdoor is introduced. During the exfiltration phase, the attacker interacts solely with the model, leveraging the backdoor and the trigger to retrieve sensitive information. This distinction is critical for understanding the attack's practicality and motivation.
> > >
> > > [1] Network Anomaly Detection: Methods, Systems and Tools, IEEE Communications Surveys & Tutorials 2014.
> > >
> > > [2] Anomaly-based network intrusion detection: Techniques, systems and challenges, Computers & Security 2009.
> > >
> > > [3] Backdoor Learning: A Survey, IEEE TNNLS 2022.
> > >
> > > [4] Backdoor Attacks to Deep Neural Networks: A Survey of the Literature, Challenges, and Future Research Directions, IEEE Access 2024.
> > >
> > > [5] BadNets: Evaluating Backdooring Attacks on Deep Neural Networks, IEEE Access 2019.
> > >
> > > [6] Extracting Training Data from Diffusion Models. USENIX 2023.

---

### Official Review · Reviewer_Cao3 · 2024-11-04

**Soundness:** 3
**Presentation:** 3
**Contribution:** 3
**Rating:** 8
**Confidence:** 3

**Summary:**

This work considers a new type of threat to data confidentiality, and proposes the use of unique trigger embeddings to enable covert extraction of private training data. Specifically, the authors introduce a new method to integrate backdoors into diffusion models, and propose novel trigger embeddings for activating these backdoors, demonstrating the potential for unauthorized data exfiltration. This is a very interesting work, and I do enjoy reading the paper.

**Strengths:**

Please see above summary.

**Weaknesses:**

It's not really a weakness, but more like a possible further discussion of possible defense mechanism.

**Questions:**

Just a question regarding the possible defense mechanism: as the authors mentioned in the paper, for this attack to work, the attackers need access to both the training data and the training process (e.g., being able to modify the loss function), which in my opinion is a very strong assumption of the attackers’ abilities. Could the authors discuss examples or scenarios where the inside attackers have this level of access and the prevalence of such access patterns in real-world ML development environments? Also, when considering the possible defense mechanism, could the authors comment on the use of traditional access control methods to limit access and make it less likely for this attack to happen? For example, is it possible to separate people who have access to the dataset and those who can access the training process?

---

> ### Author Response · Authors · 2024-11-20
>
> We appreciate the reviewer's insightful question regarding the assumptions made about the attacker's capabilities. Indeed, we acknowledge that access to both the training data and the ability to modify the training process is a relatively strong assumption as compared with other threat models, such as data poisoning alone or model theft. However, we would like to highlight several real-world scenarios where attackers can possess varying levels of access in ML development environments.
>
> 1. Full-Access Environment:
>
>     In this scenario, the attacker is a team member responsible for training a model using the dataset. The attacker can easily modify the training code to implant a backdoor into the diffusion model during the training process. This approach enables covert data exfiltration by embedding a trigger within the model, avoiding direct dataset downloads that could be detected through network monitoring.
>
> 2. Protected Environment:
>
>     The attacker has access to the training process but not to the raw training dataset, which is securely stored on a remote server. In this case, the attacker cannot directly download or inspect the images from the training dataset. However, the attacker can still inject a backdoor by recording features of each image the model encounters during training and dynamically assigning corresponding trigger embeddings to these images. Even without direct access to the dataset, the attacker can exploit training iterations to implant a backdoor covertly.
>
> 3. Secure Environment:
>
>     In this setup, the attacker does not participate directly in the training process but acts as an algorithm provider. This scenario is practical in real-world settings where an attacker can upload malicious training code to platforms like GitHub. A non-expert user might inadvertently use this code to train their models, unknowingly introducing a backdoor. Similarly, in ML marketplaces (e.g., Algorithmia), an attacker could sell a compromised algorithm that users adopt, leading to data leakage once the trained model is deployed. Given that our proposed method maintains strong benign performance, it is challenging for users to detect the presence of a backdoor.
>
> **Defense Mechanisms and Access Control**:
>
> In secure and protected environments, even if an attacker employs mechanisms to track images used during training and assigns specific triggers, strong defenses can significantly mitigate such attacks. For example, image augmentations [1,2] can introduce variations across training epochs, effectively obfuscating the identities of individual images. When these augmented images exhibit noticeable differences from their original counterparts, it becomes challenging for an attacker to rely on simple similarity metrics to consistently identify identical images. Consequently, the model may mistakenly treat augmented images from the same image as different entities, consuming the model's memory capacity and leading to a reduction in the attack success rate (ASR).
>
> [1] CutMix: Regularization Strategy to Train Strong Classifiers with Localizable Features, ICCV’19.
>
> [2] Effective Data Augmentation With Diffusion Models, ICLR’24.

---

### Comment · Area_Chair_2kp3 · 2024-11-24

Dear reviewers,

Thanks for serving as a reviewer. As the discussion period comes to a close and the authors have submitted their rebuttals, I kindly ask you to take a moment to review them and provide any final comments.

If you have already updated your comments, please disregard this message.

Thank you once again for your dedication to the OpenReview process.

Best,

Area Chair

---

### Note · Authors · 2025-03-06

I have read and agree with the venue's withdrawal policy on behalf of myself and my co-authors.

---

### Meta-Review · Area_Chair_2kp3 · 2024-12-20

**Metareview:**

The paper proposes a new threat to data extraction in diffusion models by embedding unique triggers in the model's training for extracting private training data. The topic paper studies is quite interesting and important. However, the assumed attacker's capabilities are too strong to weaken its contributions and impacts.

Strengths:
1. Good writing, easy to follow;

2. Extensive experiments and evaluations.

Weaknesses:

The assumed attacker's capabilities are too strong, making the threat model unrealistic. I agree that in some cases the attackers can have such strong capability, but at that time, poisoning may not be the most effective way. In fact, just memorizing the training caption and prompting them to DMs can also cause the training data extraction as pointed out in many papers[1]. Therefore, I think the authors should consider a much harder case, like just poisoning the training data or the released pertaining models [2].

Although the authors reply that their setting is the same as diffusion models [3, 4, 5] and data exfiltration [6, 7], they are not the same, for the former diffusion backdoors, they are trying to release the backdoor model to let users do something harmful, which means the whole developer is malicious and trying to let users perform bad. As for the data exfiltration works, the attackers are usually just a part of the developer in federal learning, in short, their attack can only access part of the data benignly but they try to attack to access the whole data. However, in the paper's scenario, the attacker can control a lot, they can already access full data but still want to use the backdoor to extract them. It is more likely to be a malicious employee in a benign company. Such a scenario is not realistic these days as restricting a single person's capability is common sense to make the whole system safe. Therefore I think the authors should re-think their work setting and try to make them more realistic.

Due to the above reason, I prefer to reject this paper.

[1] Understanding and Mitigating Copying in Diffusion Models.

[2] Privacy Backdoors: Enhancing Membership Inference through Poisoning Pre-trained Models.

[3] VillanDiffusion: A Unified Backdoor Attack Framework for Diffusion Models, NeurIPS’23.

[4] How to backdoor diffusion models?, CVPR’23.

[5] Trojdiff: Trojan attacks on diffusion models with diverse targets, CVPR’23.

[6] Stealing Your Data from Compressed Machine Learning Models, ACM DAC’20.

[7] Transpose Attack: Stealing Datasets with Bidirectional Training, NDSS Symposium’24.

**Additional Comments On Reviewer Discussion:**

The reviewers discussed whether the threat model is practical or not.

---

### Decision · Program_Chairs · 2025-01-22

Reject